# The Development of Symbolic Expressions for the Detection of Hepatitis C Patients and the Disease Progression from Blood Parameters Using Genetic Programming-Symbolic Classification Algorithm

Nikola Anđelić *,†, Ivan Lorencin †, Sandi Baressi Šegota and Zlatan Car

Faculty of Engineering, University of Rijeka, Vukovarska 58, 51000 Rijeka, Croatia
* Correspondence: nandelic@riteh.hr
† These authors contributed equally to this work.

**Abstract:** Hepatitis C is an infectious disease which is caused by the Hepatitis C virus (HCV) and the virus primarily affects the liver. Based on the publicly available dataset used in this paper the idea is to develop a mathematical equation that could be used to detect HCV patients with high accuracy based on the enzymes, proteins, and biomarker values contained in a patient's blood sample using genetic programming symbolic classification (GPSC) algorithm. Not only that, but the idea was also to obtain a mathematical equation that could detect the progress of the disease i.e., Hepatitis C, Fibrosis, and Cirrhosis using the GPSC algorithm. Since the original dataset was imbalanced (a large number of healthy patients versus a small number of Hepatitis C/Fibrosis/Cirrhosis patients) the dataset was balanced using random oversampling, SMOTE, ADSYN, and Borderline SMOTE methods. The symbolic expressions (mathematical equations) were obtained using the GPSC algorithm using a rigorous process of 5-fold cross-validation with a random hyperparameter search method which had to be developed for this problem. To evaluate each symbolic expression generated with GPSC the mean and standard deviation values of accuracy (ACC), the area under the receiver operating characteristic curve ($AUC$), precision, recall, and F1-score were obtained. In a simple binary case (healthy vs. Hepatitis C patients) the best case was achieved with a dataset balanced with the Borderline SMOTE method. The results are $\overline{ACC} \pm SD(ACC)$, $\overline{AUC} \pm SD(AUC)$, $\overline{Precision} \pm SD(Precision)$, $\overline{Recall} \pm SD(Recall)$, and $\overline{F1-score} \pm SD(F1-score)$ equal to $0.99 \pm 5.8 \times 10^{-3}$, $0.99 \pm 5.4 \times 10^{-3}$, $0.998 \pm 1.3 \times 10^{-3}$, $0.98 \pm 1.19 \times 10^{-3}$, and $0.99 \pm 5.39 \times 10^{-3}$, respectively. For the multiclass problem, OneVsRestClassifer was used in combination with GPSC 5-fold cross-validation and random hyperparameter search, and the best case was achieved with a dataset balanced with the Borderline SMOTE method. To evaluate symbolic expressions obtained in this case previous evaluation metric methods were used however for $AUC$, $Precision$, $Recall$, and $F1-score$ the macro values were computed since this method calculates metrics for each label, and find their unweighted mean value. In multiclass case the $\overline{ACC} \pm SD(ACC)$, $\overline{AUC}_{macro} \pm SD(AUC)$, $\overline{Precision}_{macro} \pm SD(Precision)$, $\overline{Recall}_{macro} \pm SD(Recall)$, and $\overline{F1-score}_{macro} \pm SD(F1-score)$ are equal to $0.934 \pm 9 \times 10^{-3}$, $0.987 \pm 1.8 \times 10^{-3}$, $0.942 \pm 6.9 \times 10^{-3}$, $0.934 \pm 7.84 \times 10^{-3}$ and $0.932 \pm 8.4 \times 10^{-3}$, respectively. For the best binary and multi-class cases, the symbolic expressions are shown and evaluated on the original dataset.

**Keywords:** ADASYN; borderline SMOTE; genetic programming-symbolic classifier; Hepatitis C; fibrosis; cirrhosis; SMOTE

## 1. Introduction

According to [1], hepatitis C is the liver tissue inflammation that is caused by the hepatitis C virus (HCV). The symptoms of those infected with HCV may be yellow discoloration of skin and eyes, poor appetite, vomiting, tiredness, abdominal pain, and diarrhea [2]. Generally, the main cause of hepatitis is hepatoviruses A, B, C, D, and E. There are other

viruses that can cause hepatitis i.e., cytomegalovirus, Epstein-Barr virus, and yellow fever virus. The other causes of hepatitis, besides viruses, include heavy alcohol use, medications, toxins, autoimmune diseases, etc. The spread of hepatoviruses is different for different types. Hepatitis A and E are spread by contaminated food and water. Hepatitis B is sexually transmitted or passed during pregnancy or childbirth from mother to child. Hepatitis C is spread through infected blood (needle sharing by intravenous drug users) while hepatitis D can only infect people with hepatitis B.

HCV belongs to the genus of the Flaviviridae family called Hepacivirus. The virus particle consists of a lipid membrane envelope that is 55–65 [nm] in diameter. The glyco-proteins E1 and E2 which take part in viral attachment and cell intrusion are attached to the envelope. Inside the envelope, the icosahedral core with a diameter of 33 to 40 [nm] is located. Inside the core, the RNA material is located. According to [3,4], HCV can cause not only hepatitis C but also liver cancer and lymphomas.

HCV can cause liver fibrosis and chirrosis. Liver fibrosis is the excessive accumulation of extracellular matrix proteins that occurs in most types of chronic liver diseases. HCV is the main cause of liver fibrosis however, alcohol abuse, and nonalcoholic steatohepatitis (NASH) are also the main causes. Liver cirrhosis (end-stage liver disease) [5] is the impaired liver function that is caused by the formation of scar tissue known as fibrosis due to damage caused by liver disease. Hepatitis C is diagnosed using two blood tests i.e., the antibody test and the PCR test. The antibody blood test is used to determine if a suspected patient has been exposed to the hepatitis C virus which is achieved by testing for the presence of antibodies to the virus. The antibodies are produced by the patient immune system and are used to fight germs. Since it takes time for the patient's immune system to produce these antibodies this test will not show a positive reaction for some months after initial exposure to the hepatitis C virus. If the antibody test is positive it indicates that the patient was exposed to HCV at some stage and this does not mean that the patient is currently infected. In order to detect if the patient is currently infected the PCR test has to be utilized. The PCR test requires a blood sample and the test will check if the virus is still present by detecting if the virus is reproducing inside the patient's body. The positive PCR test indicates that the patient's body has not fought off the virus and that the patient is currently infected.

So after a positive PCR test, it can be concluded that the patient has an active hepatitis C infection and additional blood and ultrasound test performed by specialists are required to check if the patient's liver has been damaged. The additional tests are blood tests and ultrasound scans. The blood test is used to detect liver damage or inflammation. On the other hand, ultrasound scans are used to test the patient's liver stiffness since stiffness suggests that the liver is scarred.

Over the last decade, various artificial intelligence (AI) algorithms have been used to detect Hepatitis C or Hepatitis C stage. The AI system was proposed in [6] that uses Gaussian support vector machines learning algorithm to predict the hepatitis C staging. The results of the conducted investigation with the Gaussian SVM learning algorithm achieved an accuracy of 97.9%. The K-Nearest Neighbors (KNN) and random forest have been used in [7] for the prediction of HCV in the Egyptian patient's dataset. The highest accuracy with KNN and random forest achieved are 51.05% and 54.56% in multi and binary class labels respectively. The neural networks, naive Bayes, decision tree, SVM, random forest, and Bayesian network have been used in [8] for early prediction of cirrhotic patients based on the Egyptian dataset. Among all these ML algorithms the Bayesian network algorithm achieved the highest performance (AUC = 74.8% and accuracy 68.9%). The Intelligence Hepatitis C Stage System (IHSDS) with ANN was used to predict the stage of hepatitis C in [9]. The model achieved 94.9% classification accuracy. The decision tree, genetic algorithm (GA) particle swarm optimization, and multi-linear regression models were developed and used in [10] for predictions of advanced fibrosis by combining the serum biomarkers and clinical information to develop the classification models. The results of the study showed that machine learning (ML) algorithms were able to predict advanced fibrosis in patients with AUCROC in the range 0.73–0.76 and with an accuracy of 66.3–84.4%,

respectively. The CatBoost, XGBoost, RFGini, LightGBM, Random forest (RF), and KNN have been used in [11] to detect Hepatitis C patients The result showed that of all ML modes highest accuracy (0.9593), recall (0.6667), precision (1), and F1-score (0.7867) was achieved with XGBoost algorithm. In [12] the supervised learning (decision tree, logistic regression, KNN, Extreme Gradient Boosting, Gradient Boosting Machine, Gaussian Naive Bayes, RF, Gradient Boosting, SVM), and unsupervised learning (K-means, Hierarchical clustering, DBMSCN, Gaussian Mixture, and K-means) models were used to detect the Hepatitis C virus from a dataset containing laboratory data of Hepatitis C patients and blood donors. The results showed that Logistic Regression and Gaussian Mixture models achieved the best accuracy score which is equivalent to 0.943 and the mutual information score of 0.9771, respectively. The Hepatitis C patient's outcome was investigated in [13] using classification techniques such as Logistic Regression, Decision Tree, SVM, and Naive Bayes. The results of this investigation showed that the highest accuracy (87.17%) was achieved using SVM. The ML models (Logistic Regression, Naive Bayes, Decision Tree, Random Forest, Extreme Gradient Boosting, kNN, SVM, ANN, and Ensemble methods) were built in [14] to predict the extent of fibrosis in patients with chronic Hepatitis C. Among all ML algorithms, the XGB achieved the highest accuracy (0.84), specificity (0.95), and specificity (0.73).

There is some notable research in which AI and ML algorithms have been used in the detection of liver cancer. In [15] the authors have used the support vector machines method for identifying the liver cancer tumor for ultrasound images. The results showed that using this method a classification accuracy of 96.72% was achieved. The ANN and logistic regression have been used in [16] to develop a model for predicting the development of liver cancer within 6 years of diagnosis with type II diabetes. The best results were achieved with ANN in terms of sensitivity (75.7%), specificity (75.5%), and the area under the receiver operating characteristic curve (87.3%). The ANN and classification of regression tree have been used in [17] on a dataset collected from the cancer registration database in Northern Taiwan medical center from 2004 to 2008 to predict the survival of patients with liver cancer. The best results were achieved with ANN in terms of accuracy (87%), sensitivity (88%), specificity (87%), and area under the receiver operating characteristic curve (91.5%). The ensemble method has been developed and used in [18] to predict liver cancer in patients based on DNA sequence. Initially, the Naive Bayes, (GLM), kNN, SVM and C5.0 Decision Tree have been considered as elements of the ensemble method however, the best results were achieved with the ensemble method consisting of C5.0 Decision Tree, kNN, and SVM. With this ensemble method the achieved accuracy, sensitivity, and specificity in prediction of the liver cancer are 88.4%, 88.4%, and 91.6%, respectively.

As seen from the previous literature overview various ML algorithms have been utilized to detect hepatitis C patients with relatively high classification accuracy. The problem that arises from the majority of utilized ML models is the inability to transform these models into simple mathematical equations which could be easily used for the detection of Hepatitis C patients with high classification accuracy. Generally, mathematical equations require fewer computational resources when compared to the entire ML models.

The novelty of this research is to show how to apply the genetic programming-symbolic classifier (GPSC) on a publicly available dataset [19] to obtain symbolic expressions which could detect the Hepatitis C patients and/or to detect the stage of Hepatitis C (hepatitis C/Fibrosis/cirrhosis) with high classification accuracy. Due to a large imbalance between class samples in the original dataset (a large number of healthy patients and a small number of Hepatitis C, Fibrosis), the novelty is to show how using dataset balancing methods can balance the dataset and in the end influence the classification accuracy of obtained symbolic expression. Since the original dataset is highly imbalanced only balanced variations of the original dataset will be used in this research and the best symbolic expressions will be evaluated on the original dataset. To summarize the novelty of this paper is to show the procedure of how symbolic expressions can be obtained using the GPSC algorithm and unbalanced dataset for the detection of Hepatitis C patients us-

ing the parameters (enzymes, proteins, and biomarker values) of their blood samples as input variables.

The GPSC is a variant of GP alongside symbolic regression and it is a method that evolves the randomly generated initial population that is not fit for solving a particular problem and making them fit with the application of crossover and mutation from generation to generation. Generally, GP is classified as an evolutionary algorithm, however, the process of obtaining the symbolic expression is similar to the supervised learning method i.e., the GP requires a dataset with defined input variables and targeted output variable from which symbolic expression is obtained. From the previous literature overview and the idea/novelty of this paper the following questions arise:

- is it possible to obtain symbolic expressions that can detect Hepatitis C and the progress of the disease (Hepatitis C/fibrosis/cirrhosis) from the parameters (enzymes, proteins, and biomarker values) of blood samples as input variables with high classification accuracy using GPSC algorithm?
- due to the high imbalance between class samples in the original dataset is it possible to apply different balancing methods (oversampling) to achieve a balance between class samples and used these datasets to obtain symbolic expressions using the GPSC algorithm with high classification accuracy?

This paper consists of the following sections Materials and Methods, Results, Discussion, and Conclusion. In the Materials and Methods section, the research methodology is described as well as the dataset, oversampling techniques, GPSC, random hyperparameter search, one versus rest classifier, evaluation metrics and methodology and computational resources used in this research. In the Results section, the results obtained in the case of binary and multiclass classification are presented. Then the best symbolic expressions are shown in both cases with classification accuracy shown when these symbolic expressions are applied to the original dataset. In the Discussion section, the previously shown results are discussed and finally, in the Conclusions section, the conclusions are given based on the conducted investigation as an answer to hypotheses derived in this section.

## 2. Materials and Methods

In this section, the research methodology, dataset, oversampling techniques, GPSC, random hyperaparamters search method with 5-fold cross-validation, evaluation metrics, and methodology, and computation resources used in this research, are described.

### 2.1. Research Methodology

In this research, the oversampling methods have been applied due to the fact that the original dataset has a large imbalance between healthy patients and patients with Hepatitis C. The methods that were used to oversample the minority class/classes are:

- Random Oversampling,
- Synthetic Minority Oversampling Technique (SMOTE),
- Adaptive Synthetic (ADASYN), and
- Borderline Synthetic Minority Oversampling Techniques (BorderlineSMOTE).

Each dataset variation is then splitted on train and test dataset in 70/30% ratio where train dataset is used in GPSC with random hyperparameters search and 5 fold cross validation process. The entire schematic view of research methodology process is shown in Figure 1.

As seen from Figure 1 there are two types of investigations conducted in this paper i.e., binary case and multiclass case. In the binary case, the hepatitis C, fibrosis, and cirrhosis patients are labeled with class number 1 and healthy with class number 0. Due to the large imbalance between class samples (a large number of class 0 samples vs. a small number of class 1 samples), the balancing methods (random oversampling, ADASYN, SMOTE, Borderline SMOTE) were applied which resulted in 4 balanced datasets. These datasets were used in GPSC with random hyperparameter search and 5-fold cross-validation to

obtain symbolic expressions and the best of them in terms of the highest mean and standard deviation values of accuracy ($ACC$), the area under the receiver operating characteristic curve ($AUC$), precision, recall, and F1-score are shown.

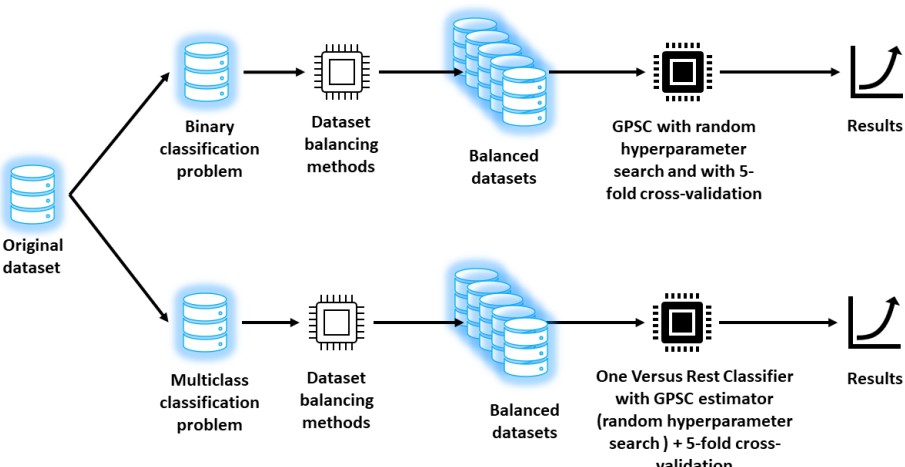

**Figure 1.** The schematic view of research methodology process.

In the multiclass classification problem shown in Figure 1, the class was labeled as 0-Healthy, 1—Hepatitis C, 2—Fibrosis, and 3—Cirrhosis. The aforementioned balancing methods were used to increase the number of samples for classes 1, 2, and 3. The 4 balanced datasets were used in GPSC with random hyperparameters search method and a 5-fold cross-validation process to obtain symbolic expressions. However, since this is a multiclass case the OneVsRestClassifer was utilized alongside GPSC. So the final output of the GPSC with OneVsRestClassifier will be the system of 4 symbolic expressions where the first symbolic expression is used to detect healthy patients, the second for Hepatitis C patients, the third for patients with fibrosis, and the fourth equation for detection of cirrhosis patients. To evaluate these symbolic expressions the mean and standard deviation values of $ACC$, $AUC_{macro}$, $Precision_{macro}$, and $F1-score_{macro}$ were used. The macro option was used since it calculates metrics for each label and finds their unweighted mean which is ideal for balanced datasets.

After the best symbolic expressions were obtained in each case the final evaluation was performed on the original imbalanced dataset to see if these symbolic expressions can detect Hepatitis C patients on real data.

*2.2. Dataset Description*

As already stated this investigation is based on a publicly available dataset which can be downloaded from Kaggle [19]. The original dataset consists of 14 variables (columns) and 615 samples (observations) of healthy patients ("blood donors") and Hepatitis C patients (categories: hepatitis C, fibrosis, cirrhosis). Each sample in the dataset has the following features Unnamed-0, Category, Age, Sex, ALP, ALB, ALT, AST, CHE, BIL, CHOL, CREA, GGT, and PROT. However, Unnamed-0 represents the patient ID./No. so it was omitted from further investigation. In the original dataset for some samples, the features were missing so these samples were omitted from the dataset. After cleaning the dataset the total number of samples is 589. In the following paragraphs, a short description of each parameter and transformation to numeric format will be given.

The age parameter represents the age of the patient whose blood sample is collected. The sex variable has two values "m" for males and "f" for females which are converted to 0 for "m" and 1 for "f". The conversion to numeric format was done so that the dataset could be used in the GPSC algorithm. The statistical information about "age" and "sex" parameters are given in Table 1.

The alkaline phosphate enzyme (ALP) [20] is a liver enzyme that is primarily found in the liver, although a smaller quantity can be found in bones. The level of ALP is measured through the ALP test and the normal range is considered from 44 to 147 [IU/L] (international units per liter), although some organizations recommend a range of 30 to 120 [IU/L] [21]. Very high values of ALP in the blood may indicate liver disease (cholestasis of pregnancy, liver cirrhosis, hepatitis, biliary atresia/stricture/obstruction due to cancer, mononucleosis) or bone disorders (bone metastasis, osteitis deformans, osteogenic sarcoma, healing fractures, Hyperparathyrodisim, hyperthyroidism, and osteomalacia). In case the ALP is way below the previously mentioned ranges it can indicate conditions such as malnutrition, zinc deficiency, magnesium deficiency, hypothyroidism, and Wilson disease.

Albumin (ALB) is a protein that is developed in the liver and the quantity in the patient's blood is measured using an albumin blood test [22]. The normal ALB range in the blood is considered from 3.4 to 5.3 [g/dL] (34 to 54 [g/L]) [23]. The low levels of ALB in the blood may indicate infections, and inflamation due to sepsis, inflammatory bowel disease, kidney disease, cirrhosis, fatty liver disease, liver cancer, or hepatitis (A/B/C). However, higher values of ALB indicate dehydration and severe diarrhea.

The alanine transaminase (ALT) enzyme is an enzyme that can be found in the liver [24]. According to [25], the normal range of ALT is between 4 and 36 [U/L], although the normal value range may vary slightly among different laboratories. The increased ALT value may indicate a sign of liver disease i.e., scarring of the liver (cirrhosis) death of liver tissue, swollen and inflamed liver (hepatitis), hemochromatosis, fatty liver, liver ischemia, liver tumor or cancer, mononucleosis, and a swollen and inflamed pancreas.

Cholinesterase (CHE) is an enzyme that helps the proper functioning of the nervous system [26]. Serum cholinesterase is a blood test that looks at the concentration of two substances acetylcholinesterase and pseudocholinesterase. Acetylcholinesterase can be found in nerve tissue and red blood cells while pseudocholinesterase is found in the liver. The normal pseudocholinesterase values are in the range 8-18 [U/L] or 8-18 [kU/L] [27]. The decreased pseudocholinesterase levels may indicate chronic infection, malnutrition, heart attack, liver damage, metastasis, and inflammation that accompanies some diseases.

Bilirubin (BIL) is a red/orange compound that occurs in the normal catabolic pathway that breaks down heme in vertebrates. This process is necessary to clear the waste products that arise from the destruction of aged or abnormal red blood cells [28]. The normal range of bilirubin for adults is in the range of 1.2 [mg/dL] [29]. High bilirubin may indicate anemia, cirrhosis, a reaction to a blood transfusion, Gilbert syndrome, viral hepatitis, reaction to drugs, alcoholic liver disease, and gallstones.

The cholesterol (CHOL) level is one of the parameters measured in blood samples [30]. All the CHOL has been created in the liver that the patient's body needs. However, additional CHOL arrives from foods the patient consumes. The normal level of total CHOL is less than 200 [mg/dL] (5.17 [mmol/L]). The borderline high level of total CHOL is in the range of 200–230 [mg/dL] (5.17–6.18 [mmol/L]). All values higher than 240 [mg/dL] (6.21 [mmol/L]) are considered high total CHOL values [31].

The creatinine (CREA) [32] number obtained from blood sample analysis indicates how well the patients' kidneys are working. The normal range of CREA is between 0.7 to 1.3 [mg/dL] for men and 0.6 to 1.1 [mg/dL] for women [33]. The values above predefined ranges for men and women may indicate a blocked urinary tract, kidney problems (damage, failure, infection, reduced blood flow), loss of body fluid, and muscle problems.

The gamma-glutamyltransferase (GGT) [34] is an enzyme that is mainly found in a patient's liver. By measuring the GGT concentration in a patient's blood its activity is being measured. The normal range of GGT in the blood is 5–40 [U/L], according to [35]. Since higher levels of GGT in a blood sample can indicate liver damage/disease the GGT is used as a diagnostic marker for liver disease. The latent elevations of GGT can be recorded in patients with chronic viral hepatitis infections which often take 12 months or more to present.

The total protein (PROT) [36] is the parameter that represents the total amount of two classes of proteins found in blood and these are ALB and globulins. The globulins are a group of proteins in the blood and are made in the liver by the patient's immune system. The normal range of PROT is 60 to 83 [g/L] [37]. The higher than normal levels of PROT may indicate chronic inflammation or infection (HIV, hepatitis B/C), multiple myeloma, and Waldenstrom disease. However, lower-than-normal levels of PRT may indicate bleeding, burns, glomerulonephritis, liver disease, malabsorption, malnutrition, nephrotic syndrome, and protein-losing enteropathy. The initial statistical analysis of the dataset is shown in Table 1.

**Table 1.** Results of statistical analysis of the dataset with definition of input and output dataset variables.

| Variable Name | GPSC Variable Representation | Count | Mean | Std | Range |
|---|---|---|---|---|---|
| Age | $X_0$ | | 47.41766 | 9.931334 | 23–77 |
| Sex | $X_1$ | | 0.616299 | 0.4867 | 0–1 |
| ALB | $X_2$ | | 41.62428 | 5.761794 | 14.9–82.2 |
| ALP | $X_3$ | | 68.12309 | 25.92107 | 11.3–416.6 |
| ALT | $X_4$ | | 26.57538 | 20.86312 | 0.9–325.3 |
| AST | $X_5$ | | 33.77284 | 32.86687 | 10.6–324 |
| BIL | $X_6$ | | 11.01817 | 17.40657 | 0.8–209 |
| CHE | $X_7$ | 589 | 8.203633 | 2.191073 | 1.42–16.41 |
| CHOL | $X_8$ | | 5.391341 | 1.128954 | 1.43–9.67 |
| CREA | $X_9$ | | 81.6691 | 50.69699 | 8–1079.1 |
| GGT | $X_{10}$ | | 38.19847 | 54.30241 | 4.5–650.9 |
| PROT | $X_{11}$ | | 71.89015 | 5.348883 | 44.8–86.5 |
| Category Binary | $y$ | | 0.095 | 0.293 | 0–1 |
| Category Multivariate | | | 0.196944 | 0.666439 | 0-3 |

The correlation analysis can be used as an initial indicator of how well are variable correlated. If the variables do not have any correlation with each other it is hardly feasible to assume that the ML model will be able to establish the correlation between investigated variables. The correlation can be described as the connection between the input and output variables. In this case, Pearson's correlation analysis was used and the correlation value can be in the range from −1.0 up to 1.0. If the correlation value is equal to −1.0 this means that if the value of the input variable rises the value of the output variable would drop and vice versa. In case Pearson's correlation is equal to 1.0 this means that if the value of the input variable increases the value of the output variable will also increase. The best correlation ranges are from −1.0 to −0.5 and from 0.5 to 1.0. The worst correlation range is from −0.5 to 0.5. Possibly the worst correlation value is 0 which means that if the value of the input variable increases/decreases it will absolutely not have any effect on the output variable value. The result of Pearson's correlation analysis performed on the original dataset is shown in Figure 2.

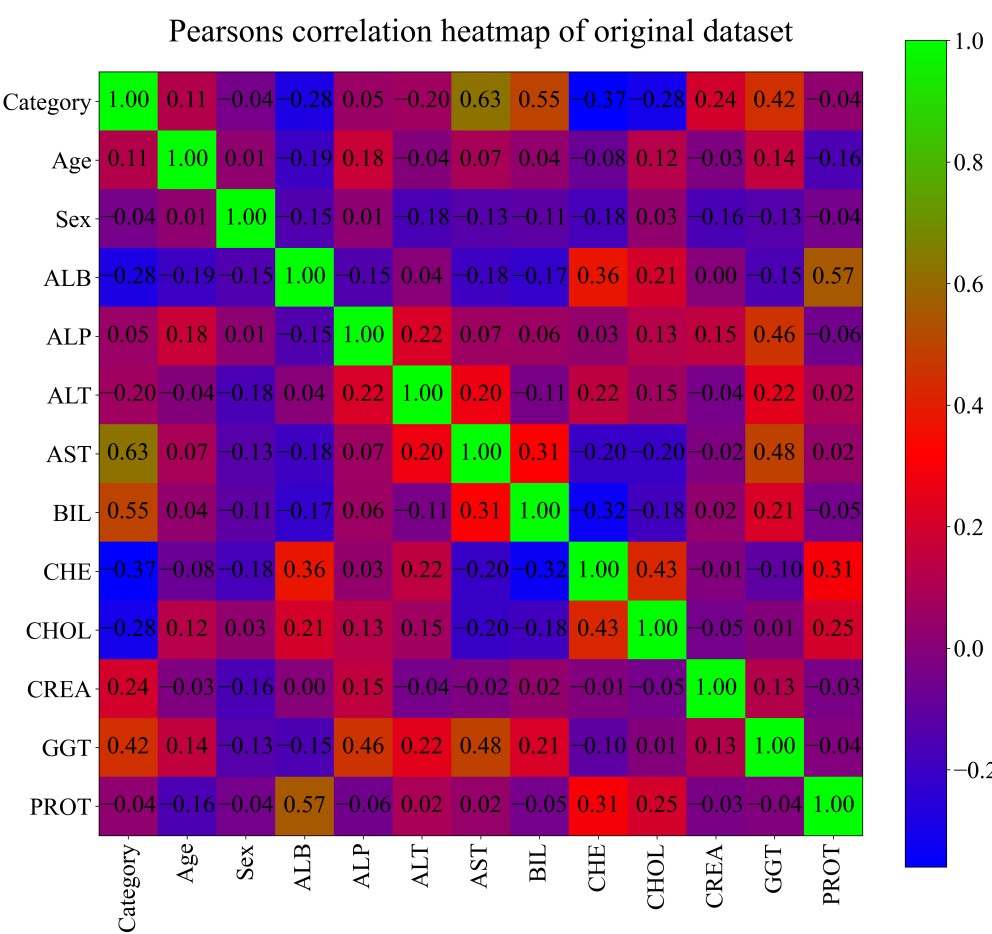

**Figure 2.** The Pearsons correlation heatmap of original dataset variables.

It can be seen from Figure 2 that the highest correlation is achieved between Category (target variable) and the input variables AST (0.63) and BIL (0.55). However, in this investigation, all input variables will be used in GPSC to develop symbolic expressions.

As already stated in this paper two different cases were investigated i.e., the binary and the multi-class problem. Initially, the dataset contains four classes "0—Blood Test" (renamed to healthy), "1—Hepatitis C", "2—Fibrosis", and "3—Cirrhosis". The classes were renamed with numbers 0, 1, 2, and 3, respectively. In the case of the binary problem classes, 2 and 3 are joined with class 1. The number of samples per class and for binary/multiclass problems is shown in Figure 3.

As seen from both subfigures in Figure 3 both cases have under-sampled classes. In the binary case, Hepatitis C is the under-sampled class while in the multiclass case the Hepatitis C, Fibrosis, and Cirrhosis class are all undersampled. To show the distribution of class samples the number of dimensions had to be reduced. The number of input variables in this investigation is 12 (12-dimensional space) and to visualize all samples for each class in 2-dimensional space the number of dimensions had to be reduced from 12 to 2-D.

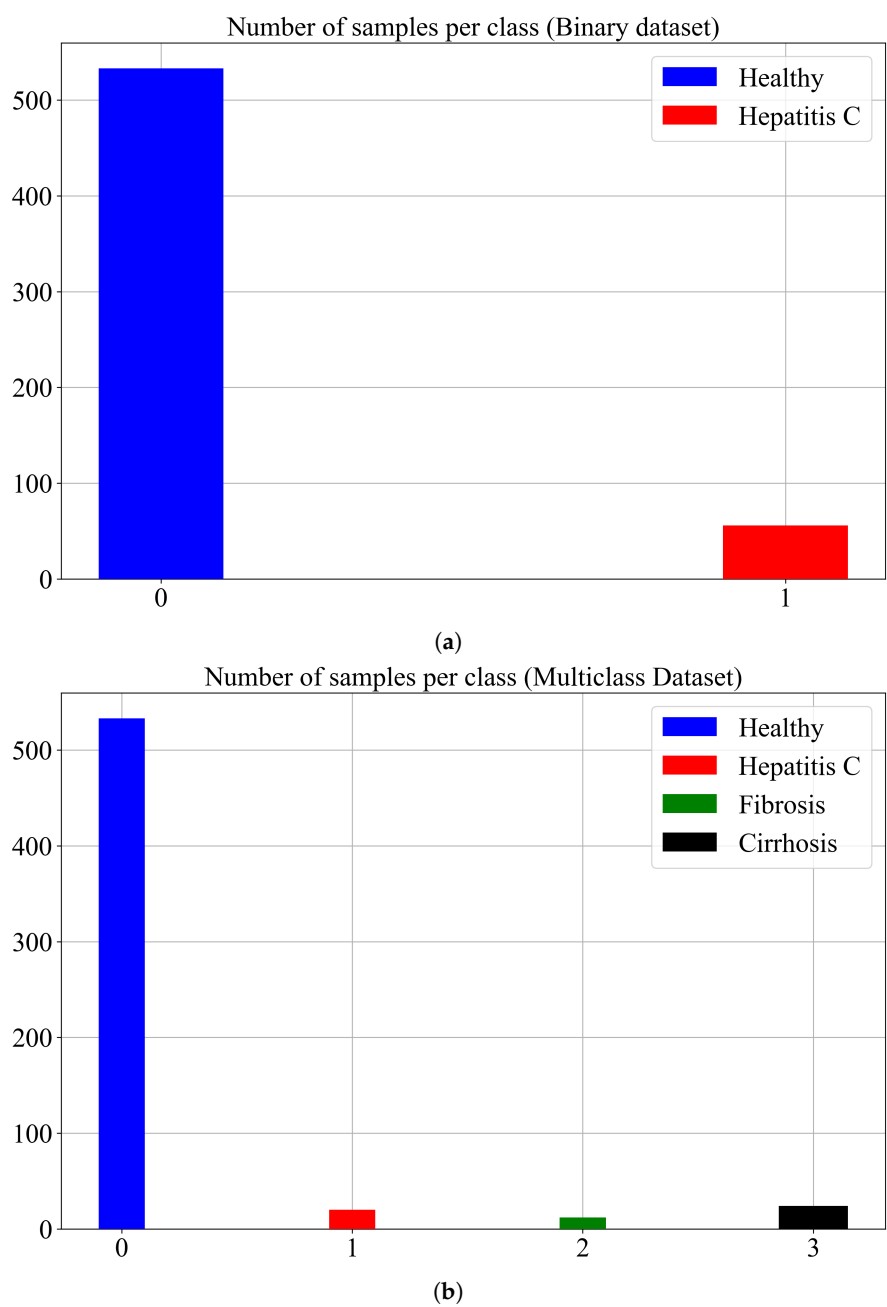

**Figure 3.** The number of samples per class in binary dataset and multiclass dataset. (**a**) Binary case; (**b**) Multiclass case.

Kernel PCA

The Kernel Principal Component Analysis (Kernel PCA), according to [38] is the nonlinear form of PCA that achieves non-linear dimensionality reduction through the use of kernels. In this paper, the Kernel PCA was used to reduce the dataset which has 12 input variables (12-dimensional space), and using the kernel function reduces it to 2-dimensional space. The reason for using Kernel PCA was to graphically show all class samples in 2-dimensional space and to visualize synthetically generated dataset samples with the application of random oversampling, SMOTE, BorderlineSMOTE, and ADASYN oversampling methods. Through trial and error, the best graphical representation was achieved using the radial basis function (RBF) kernel function. The Kernel PCA method [38] can be summarized in 4 steps i.e., construct the kernel matrix from the initial dataset, compute Gram matrix, use the Gram matrix to calculate the vectors $\mathbf{a}_i$ and using them

compute the kernel principal components. In Figure 4 the results of Kernel PCA applied on the original binary and multiclass dataset are shown in form of scatter plots with two kernel principal components ($KPCA_1$, $KPCA_2$).

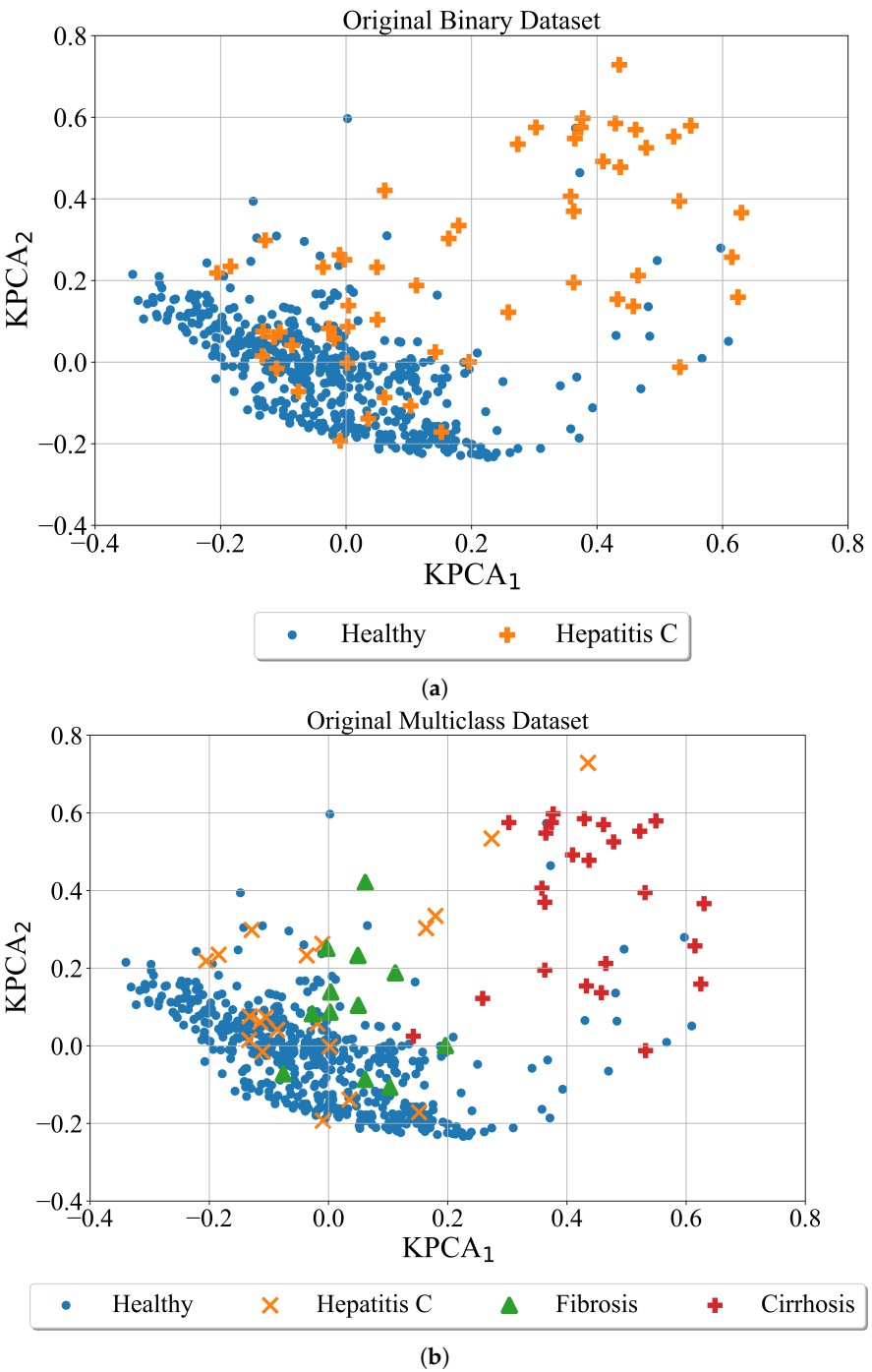

**Figure 4.** The results of Kernel PCA application on original binary and multiclass datasets. (**a**) Binary scatter plot; (**b**) Multiclass scatter plot.

As seen from Figure 4 the Hepatitis C, fibrosis, and cirrhosis patients overlap with Healthy patients. One of the reasons why these classes overlap is that some of the blood parameters of unhealthy patients are the same or in similar range to those of healthy patients.

### *2.3. Oversampling Methods*

The oversampling methods are used to balance the number of samples of dataset classes. In the case of the imbalanced dataset with two classes, the class with a lower number of samples is called the minority class, while the other is called the majority class. So with the application of oversampling techniques, the idea is to oversample the number of minority class samples to match the number of samples of the majority class. In the case of multiple classes, the idea is to oversample the number of samples of minority classes to match the sample number of the majority class. As already stated in this research Random Oversampling, SMOTE, ADASYN, and Borderline SMOTE methods have been used to balance the original dataset.

#### 2.3.1. Random Oversampling

Random oversampling is a naive strategy to generate new samples when compared to other used methods. In an imbalanced dataset with two classes, the majority class is a class with a larger number of samples while the minority class is a class with a smaller number of samples. In this method, the new samples of the minority class are generated by random sampling with the replacement of the currently available samples. The random oversampling technique was applied to both datasets i.e., binary and multiclass datasets. The results of the application of random oversampling to both datasets are shown in Figure 5. However, it should be noted that for better visualization the Kernel PCA method was applied after random oversampling of the dataset.

As seen from Figure 5 visually the number of points did not increased. The graphs are almost identical to the scatter plots shown in previous Figure 4. The main problem with random oversampling method is that number of samples in undersampled classes i.e., minority classes are oversampled by randomly choosing and copying the samples from minority class. However, these are the same samples that already exist in the dataset. So the number of samples of minority class is matched to majority class but the samples in the minority class repeat.

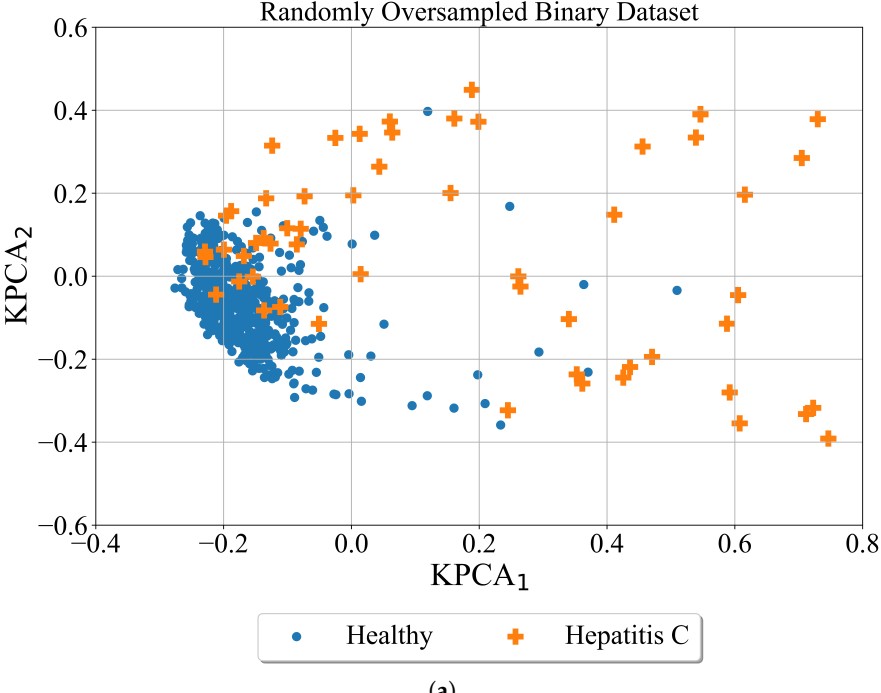

(**a**)

**Figure 5.** *Cont.*

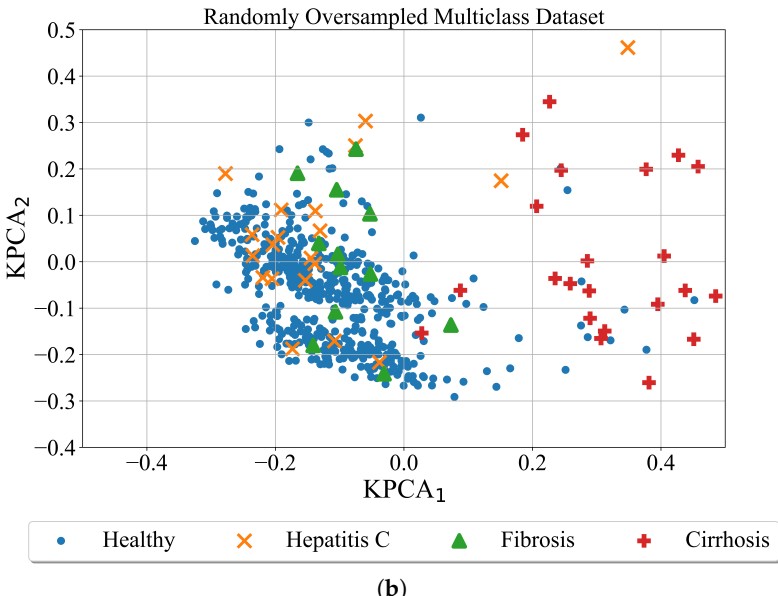

(**b**)

**Figure 5.** The results of random oversampling technique application on binary and multiclass dataset. (**a**) Binary scatter plot after random oversampling; (**b**) Multiclass scatter plot after random oversampling.

### 2.3.2. SMOTE

The Synthetic Minority Oversampling Technique (SMOTE) is an algorithm which is used to oversample the minority class or multiple classes by creating synthetic samples based on real data. The algorithm execution starts by taking a difference between the sample and its nearest neighbor. Then the difference is multiplied by a random number in 0 to 1 range and added to the sample under consideration. By doing so the random point is created along the line segment between two specific features.

After application of the SMOTE algorithm the Kernel PCA was applied just for visualization purposes to get better perspective of synthetically generated samples using SMOTE algorithm. The binary and multiclass datasets balanced with SMOTE are shown in Figure 6.

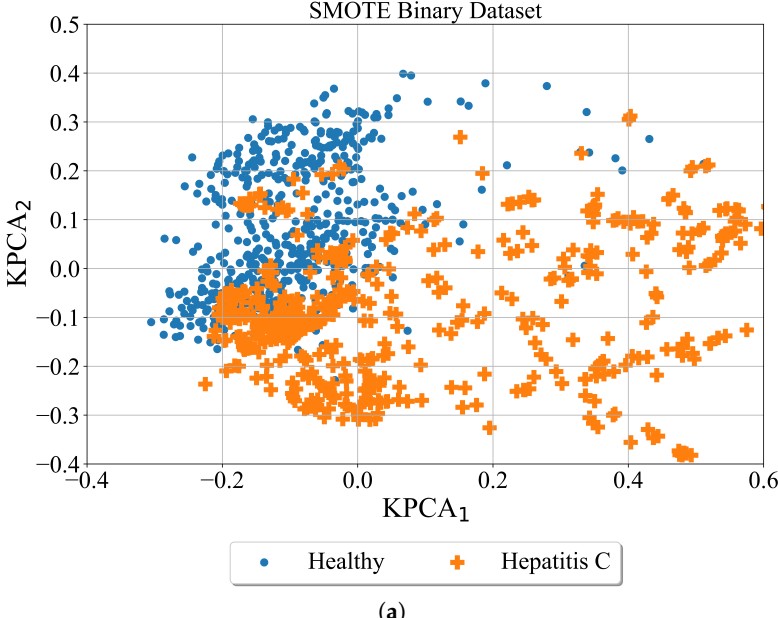

(**a**)

**Figure 6.** *Cont.*

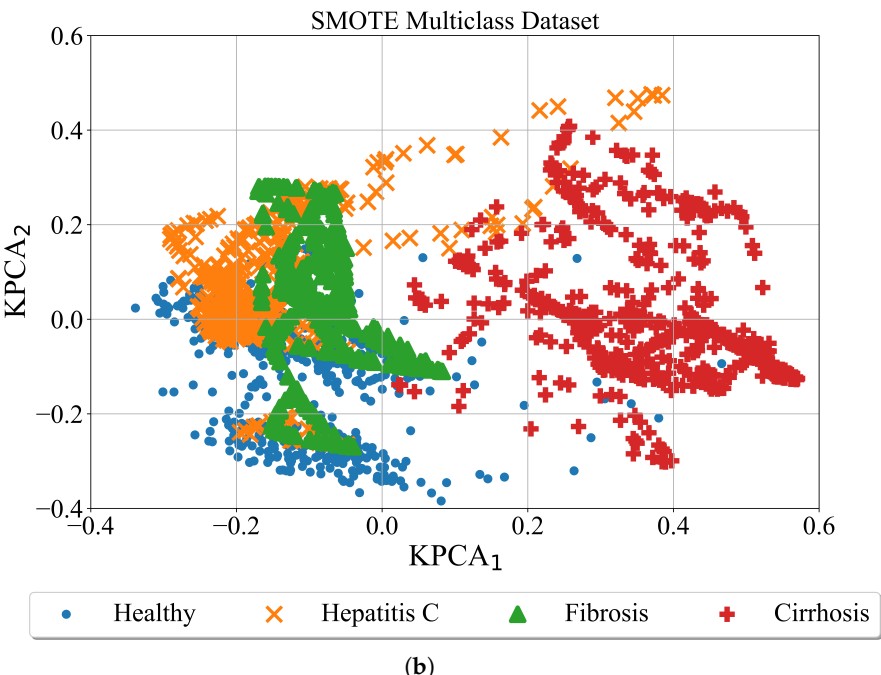

**Figure 6.** The results of SMOTE oversampling technique application on binary and multiclass dataset. (**a**) SMOTE oversampled Binary Dataset; (**b**) SMOTE Oversampled Multiclass Dataset.

In Figure 6 the results of SMOTE over-sampling technique to both binary and multiclass datasets are shown. As seen in the binary case the minority class (Hepatitis-C) is oversampled. The region of this class is now much larger although both classes overlap. In the case of the multiclass problem, three minority classes (Hepatitis C, fibrosis, and cirrhosis) are oversampled. The number of samples is much larger than in binary case and the overlapping occurs between all 4 classes.

### 2.3.3. ADASYN

The Adaptive Synthetic (ADASYN) algorithm can be described as an upgraded version of SMOTE algorithm. The initial dataset consist of $m$ samples that can generally be written as:

$$\{\mathbf{x}_i, y_i\}, i = 1, \ldots, m, \tag{1}$$

where $\mathbf{x}_i$ and $y_i$ represent an instance in the n-dimensional feature space $\mathbf{X}$ and class identity label associated with $x_i$. The total number of dataset samples is grouped based on classes where $S_s$ represents the minority class samples and $S_l$ represents the majority class samples. The conditions for the definition of minority and majority class samples are

- the number of minority samples must be less or equal to the number of majority samples,
- the sum of the minority and majority number of samples must be equal to the total number of samples.

The algorithm execution begins by inspecting the degree of class imbalance using the expression:

$$d = \frac{S_s}{S_l}. \tag{2}$$

The degree of class imbalance can be in the range of 0 to 1. The value of d is compared with a preset threshold for the maximum tolerated degree of class imbalance ratio. If the value is below the threshold the ADASYN algorithm will be applied. The next step is to calculate the number of synthetic data samples that have to be generated in the case of the minority class. To calculate the number of samples the following expression is utilized:

$$G = (S_l - S_s) \times \beta, \tag{3}$$

In the previous equation, the $\beta$ is the parameter used to define the balance level after the generation of synthetic data. The value of this parameter can be in the range from 0 to 1 and if $\beta = 1$ this means that the dataset is fully balanced after the generalization process.

The K nearest neighbors are found for each dataset minority class sample based on Euclidean distance in n-dimensional space and calculate the ratio between the number of examples $\delta_i$ in the $K$ nearest neighbors of $x_i$ that are from majority class using the expression:

$$r_i = \frac{\Delta_i}{K}, \quad i = 1, \dots, S_s. \tag{4}$$

The range of $r_i$ is between 0 and 1. After $r_i$ is obtained it has to be normalized so that the density distribution is equal to 1. The normalization is done using the expression:

$$\hat{r}_i = \frac{r_i}{\sum_{i=1}^{S_s} r_i} \tag{5}$$

The next step is to calculate the number of synthetic data samples that have to be generated for each minority sample:

$$g_i = \hat{r}_i \times G \tag{6}$$

Finally, for each minority class data sample generate synthetic data samples in following steps in range from 1 to $g_i$.

- Randomly choose one data sample $x_{zi}$ from the K nearest neighbors for data $x_i$
- Generate the synthetic data sample using the expression:

$$s_i = x_i + (x_{zi} - x_i) \times \lambda \tag{7}$$

where $(x_{zi} - x_i)$ is the difference vector in n dimensional space and $\lambda$ a random number between 0 and 1.

After the minority classes are oversampled in both cases using the ADASYN method the kernel PCA was utilized for better dataset visualization. The results are shown in Figure 7.

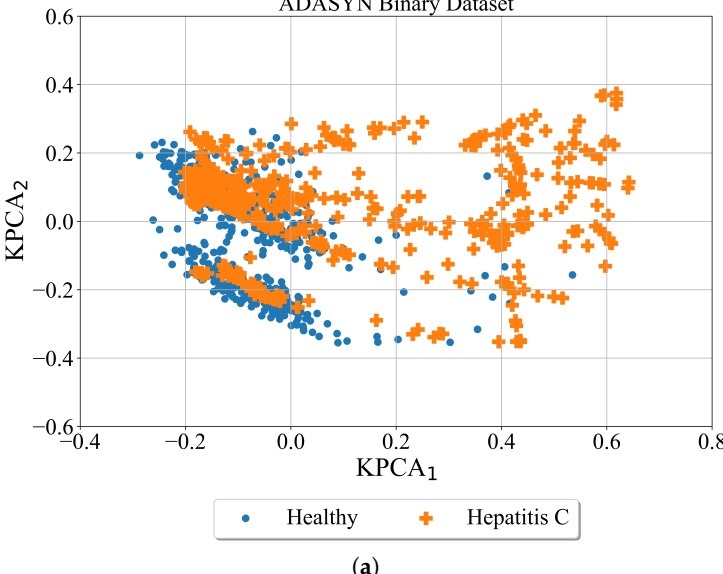

(a)

**Figure 7.** *Cont.*

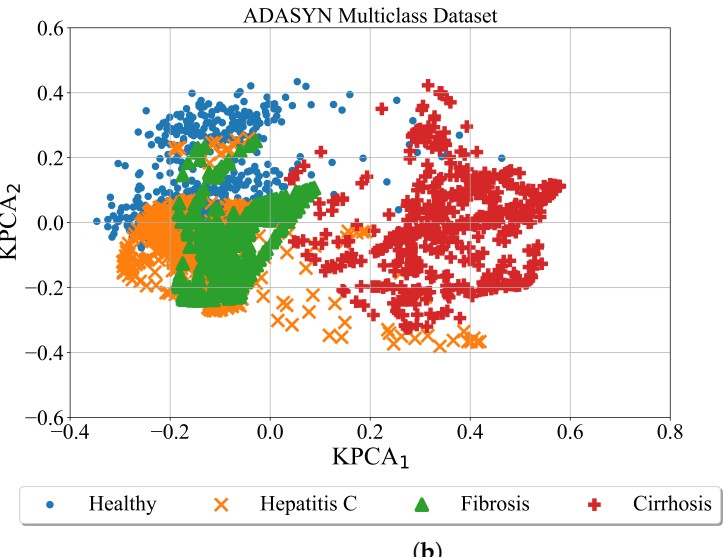

(**b**)

**Figure 7.** The results of ADASYN oversampling technique application on binary and multiclass dataset. (**a**) ADASYN oversampled Binary Dataset; (**b**) ADASYN Oversampled Multiclass Dataset.

In case of ADASYN (Figure 7) the minority classes in both cases are oversampled. However, the areas of class samples overlap. This is especially evident in multiclass case where samples of healthy patients overlap with samples of Hepatitis C and Fibrosis patients. The samples of cirrhosis patients slightly overlap with healthy, hepatitis C and fibrosis patients.

### 2.3.4. BorderlineSMOTE

The Borderline SMOTE is a variant of SMOTE algorithm in which borderline samples between two classes are detected and used to generate new synthetic samples.

For every sample in the minority class, the $m$ nearest neighbors are calculated to form the whole training set.

After the minority classes in the binary and multiclass datasets are oversampled using the Borderline SMOTE method the kernel PCA was used for better visualization of obtained results. The Binary and Multiclass datasets oversampled with the Broderline SMOTE method are shown in Figure 8.

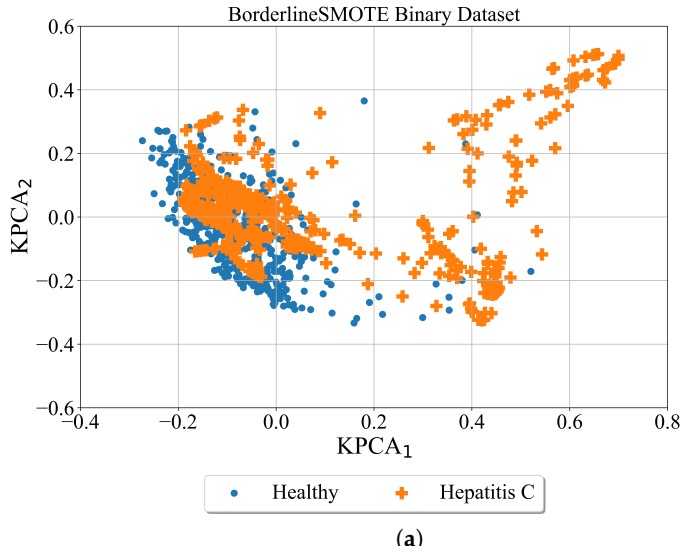

(**a**)

**Figure 8.** *Cont.*

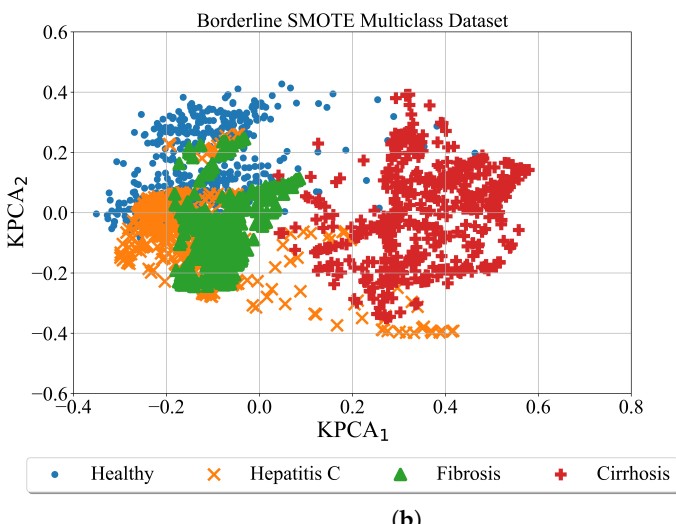

**(b)**

**Figure 8.** The results of Borderline SMOTE oversampling technique application on binary and multiclass dataset. (**a**) Borderline SMOTE oversampled Binary Dataset; (**b**) Borderline SMOTE Oversampled Multiclass Dataset.

As seen from Figure 8 after balancing the dataset with the Borderline SMOTE method the samples of each class are more condensed. Although the class samples overlap which is evident in both cases (binary and multiclass).

### 2.4. Genetic Programming—Symbolic Classifier

The genetic programming symbolic classifier (GPSC) is a method that is used for obtaining the symbolic expression that can classify the target variable with high accuracy. The algorithm begins execution by randomly creating the initial population. In GP the initial population is built using randomly selected elements from the primitive set that contains variables, constants, and mathematical functions. The input variables are defined based on the number of dataset input variables. The range of constant values is defined using GP hyperparameter constant_range and types of mathematical functions are defined with function_set hyperparameter. The mathematical functions used in this investigation are addition, subtraction, multiplication, division, minimum, maximum, sine, cosine, tangent, square root, cube root, natural logarithm, logarithm with base 2 and 10, and absolute value. The method used to create the initial population is ramped half-and-half which means that half of the initial population is created using full method [39] and the other half is created using grow method [40]. The term ramped means that the depth of initial population members is in a specific range. The depth range in GP is defined using init_depth hyperparameter.

After the initial population is created the members have to be evaluated. Each population member goes through the Sigmoid decision function and the difference between obtained output and targeted output for each sample is calculated using log_loss metric a.k.a. fitness function. The evaluated members are then randomly selected and the number of members that will compete to become the parents of the next generation is defined using tournament_size hyperparameter. The members that compete in tournament selection are randomly chosen and compared. The population member that has the lowest value of the fitness function will become the parent on which genetic operations will be performed to create offspring for the next generation.

In GPSC a total of four different genetic operations were used i.e., crossover, subtree mutation, hoist mutation, and point mutation. The crossover requires two winners of tournament selection and random subtrees on both winners are randomly selected. Then the random subtree from the second tournament winner is used and it replaces the subtree on the first tournament winner to form the offspring of the next generation. The subtree

mutation process takes only one winner of tournament selection and a random subtree on that winner is selected which is replaced with a randomly generated subtree from elements in the primitive set to form the offspring of the next generation. The hoist mutation takes the winner of the tournament selection and a random subtree is selected. Inside the randomly selected subtree, another tree is randomly selected which is then hoisted into the original tree to form offspring of the next generation. The point mutation also takes the winner of the tournament selection and randomly selects nodes on that winner. The randomly selected variables and constant nodes are replaced with elements from the primitive set. Functions are replaced with randomly selected functions however the arity of the original function must match the arity of the newly selected function. The crossover and mutation hyperparameters responsible for genetic operators are p_crossover, p_sutbree_mutation, p_hoist_mutation, and p_point_mutation. The sum of all operators has to be equal to 1, if not the population members will enter the next generation without any improvement.

The termination criteria are responsible for stopping the GPSC execution. Otherwise, the execution will go indefinitely. There are two termination criteria used i.e., stopping criteria and a maximum number of generations. The stopping criteria hyperparameter is the predefined lowest value of the fitness function which if achieved by one of the population member will terminate the execution of the algorithm. The other hyperparameter is the maximum number of generations and when the GPSC reach this number of generation it will terminate the execution.

The parsimony coefficient hyperparameter is responsible for preventing the occurrence of the bloat phenomenon. Sometimes during the GP execution, the size of population members can grow in size (depth and length) without any benefit to lowering the fitness value. The parsimony coefficient is a very useful tool since it penalizes large population members by multiplying the fitness value with the parsimony coefficient. The predefined ranges of hyperparameters used in binary and multiclass cases are listed in Table 2.

**Table 2.** The list of predefined ranges of GPSC hyperparameters used in random hyperparameter search method.

| GPSC Hyperparameter Name | Lower Value | Upper Value |
| --- | --- | --- |
| Population size | 100 | 500 |
| Number of generations | 100 | 250 |
| Tournament Size | 10 | 100 |
| Crossover | 0.001 | 1 |
| Subtree mutation | 0.001 | 1 |
| Hoist Mutation | 0.001 | 1 |
| Point Mutation | 0.001 | 1 |
| Stopping Criteria | $1 \times 10^{-6}$ | $1 \times 10^{-3}$ |
| Maximum samples | 0.6 | 1 |
| Constant Range | $-10,000$ | $10,000$ |
| Parsimony Coefficient | $1 \times 10^{-5}$ | $1 \times 10^{-4}$ |

*2.5. Random Hyperpameter Grid Search with 5-Fold Cross-Validation*

Every investigation conducted in this paper using GPSC was done using a random hyperparameter search method with 5-fold cross-validation. The random hyper-parameter search method was developed and values were randomly selected before each GPSC execution. The predefined ranges of each GPSC hyperparameter from which hyperparameters were randomly selected in each execution of GPSC are given in Table 2.

The dataset was initially divided into train/test datasets in the ratio of 70:30. The 70% dataset was used to train the symbolic classifier using 5-fold cross-validation. After this process is done the evaluation metric values are computed and if the mean values of *ACC*, *AUC*, *Precision*, *Recall*, and *F*1 − *score* are larger than the predefined values the final process using the classic train test will be performed using the same hyperparameters as in the case of 5-fold cross-validation. In case the mean values of evaluation metrics of

5-fold cross-validation are lower than the predefined values the process is repeated again i.e., the random hyperparameters are again randomly selected. The schematic view of the random hyperparameter search method with 5-fold cross-validation is shown in Figure 9.

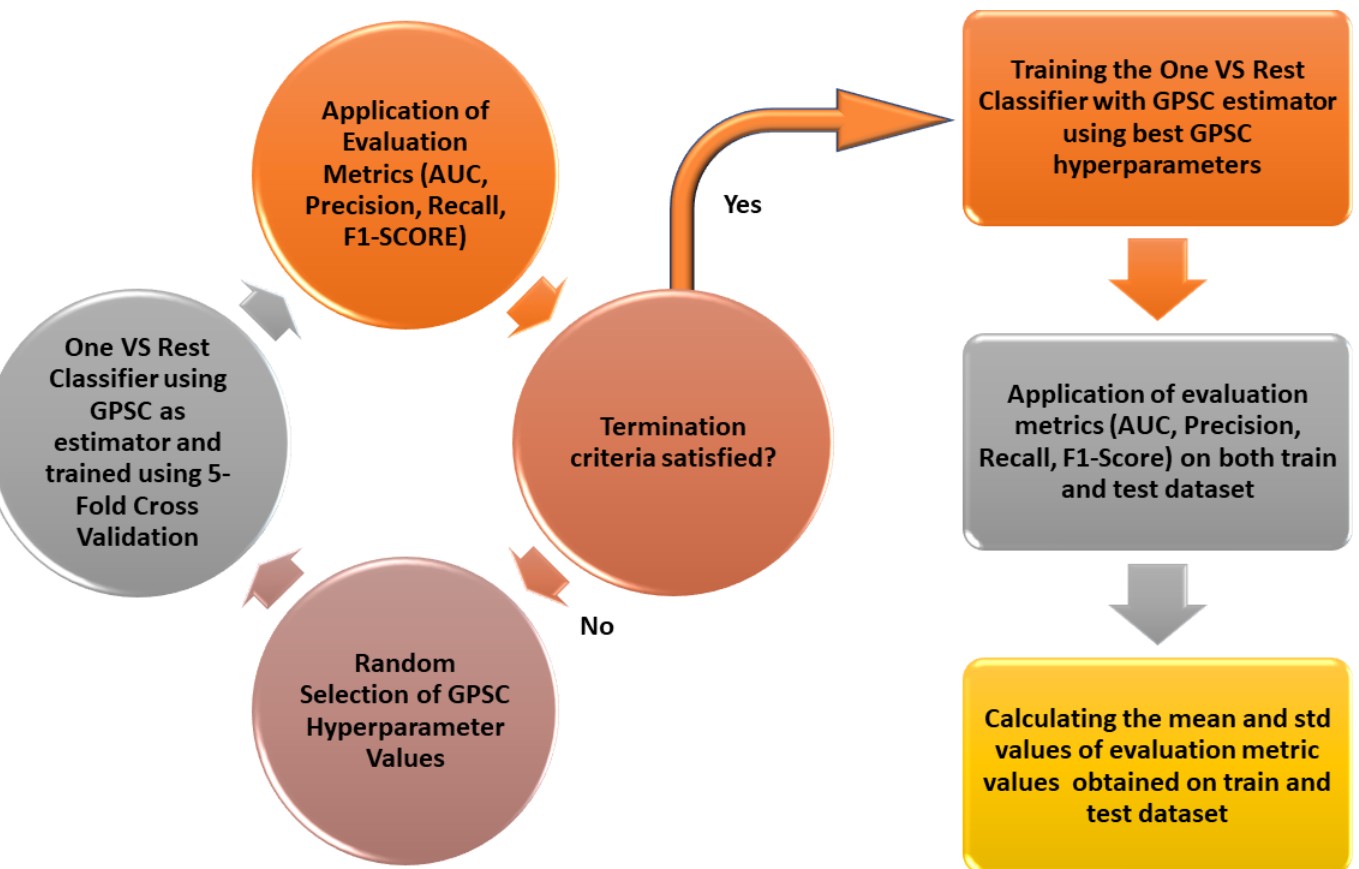

**Figure 9.** The procedure of performing GPSC with random hyperparameter search with 5-fold cross-validation.

### 2.6. One vs. Rest Classifier

In the case of a multi-class problem, the symbolic expression obtained with the GPSC algorithm must correctly classify to one of the following outputs 0—Healthy, 1—Hepatitis, 2—Fibrosis, and 3—Cirrhosis. To do that the One vs. Rest Classifier was used. However, when One vs. Rest Classifier was used with GPSC it did not produce the symbolic expression and there was not any way to access the estimator inside the OneVSRestClassifier function. To overcome this problem the One vs. Rest Classifier had to be built from scratch for GPSC to build the system. In the case of 4 different classes, the dataset had to be slightly modified i.e., in the case of detecting healthy patients from the entire dataset the healthy class is labeled as 1, and the remaining three (1—Hepatitis, 2—Fibrosis, and 3—Cirrhosis) as 0. The following dataset modification including the previously described case are:

- 0—Healthy labeled as 1 vs. (1—Hepatitis, 2—Fibrosis, 3—Cirrhosis) labeled as 0,
- 1—Hepatitis labeled as 1 vs. (0-Healthy, 2—Fibrosis, 3—Cirrhosis) labeled as 0,
- 2—Fibrosis labeled as 1 vs. (0-Healthy, 1—Hepatitis, 3—Cirrhosis) labeled as 0, and
- 3—Cirrhosis labeled as 1 vs. (0-Healthy, 2—Fibrosis, 3—Cirrhosis) labeled as 0.

After the best symbolic symbolic expressions are obtained for each case the multi-class detection system will consist of 4-different symbolic expressions. The result of each symbolic expression goes through the Sigmoid decision function and the end result is multiplied by the real class number. For example, the third symbolic expression detects 2—Fibrosis patients from the dataset. The output of the Sigmoid function can be 0 or 1. If it

is 1 then the system has detected the patient from 2—Fibrosis and to obtain the real dataset class then the output of the Sigmoid function is multiplied by 2.

### 2.7. Evaluation Metrics and Methodology

To evaluate each symbolic expression after it was obtained with the GPSC algorithm is to calculate the output values with symbolic expression by providing the inputs of train or test dataset. Then these output values go as inputs into the Sigmoid decision function and the calculated output is compared with real dataset output value to calculate precision, recall, F1-score, and of course Area under curve (AUC) value.

#### 2.7.1. Evaluation Metrics

In a classification of a specific dataset four basic combinations of actual data categories and assigned categories, are:

- true positives (TP)—correct positive assignments,
- true negatives (TN)—correct negatives,
- false positives (FP)—incorrect positive assignments,
- false negatives (FN)—incorrect negative assignments.

The accuracy score [41] can be described as a fraction of predictions that ML model got right. In binary classification the accuracy is calculated using the expression:

$$ACC = \frac{TP + TN}{TP + TN + FP + FN}. \tag{8}$$

The *AUC* score [42] computes the area under the receiver operating characteristic (ROC) curve. By computing the area under the roc curve, the curve information is summarized in one number.

According to [43], the precision is the ability of the classifier not to label as positive a sample that is negative. The precision score is the ratio between TP and the sum of TP and FP which can be written as:

$$Precision = \frac{TP}{TP + FP} \tag{9}$$

Recall [43] is the ability of the classifier to find all the positive samples. The recall is the ratio between TP and the sum of TP and FN and can be written as:

$$Recall = \frac{TP}{TP + FN} \tag{10}$$

F1-score [44] is the harmonic mean of the precision and recall and can be written as:

$$F1 - Score = \frac{2 \cdot precision \cdot recall}{precision + recall} = \frac{2TP}{2TP + FP + FN} \tag{11}$$

The majority of classification metrics are by default defined for the binary case problems. To expand these metrics to multiclass problems (Datasets), a few additional techniques have to be introduced. The multiclass problems can be broken down into a sequence of binary problems using One-vs-One (OVO) or One-vs-Rest (OVR). In this paper, the OVR was used to obtain a system of symbolic expressions for the detection of patients with HCV and their stage of the disease. Using OVR the multiclass problem is broken down into a series of binary tasks for each class in the target variable. In this case, we have 4 classes (healthy, hepatitis C, fibrosis, and cirrhosis) that are binarized to four tasks using OVR:

- task 1: healthy versus (hepatitis C, fibrosis, cirrhosis),
- task 2: hepatitis C versus (healthy, fibrosis, cirrhosis),
- task 3: fibrosis versus (healthy, hepatitis C, cirrhosis), and
- task 4: cirrhosis versus (healthy, hepatitis C, fibrosis).

To evaluate the obtained symbolic expressions in multiclass case (OneVsRestClassifier) the same evaluation metrics were used as in previous case however, the macro averaging

method was used. The macromethod calculates metrics for each label, and find their unweighted mean.

### 2.7.2. Evaluation Methodology

The process of evaluating symbolic expressions is the same in the binary case and in the multiclass case. The process starts by randomly selecting hyperparameter values of the GPSC algorithm. Then the 5-fold cross-validation is performed on the train part of the dataset (70% of the dataset). After each fold, the obtained symbolic expressions were evaluated i.e., the ACC, AUC, precision, recall, and F1-score are determined. After the process of 5-fold cross-validation is completed the mean values of the aforementioned metrics are determined. The next step is to apply "termination criteria" which is basically the condition that states if all evaluation metric values are higher than 0.9 then the final training/testing process can occur. If one of the metric values is below 0.9 the process is repeated with the random selection of new hyperparameters.

In the final stage after the obtained evaluation metric values passed the termination criteria test i.e., all mean values are greater than 0.99 the training and testing are performed with GPSC. The training process is performed using GPSC on 70% of the dataset with the same hyperparameters used in the 5-fold cross-validation process. After the training process is complete the evaluation metric values were obtained on the train and test dataset and mean values and standard deviation values were obtained.

### 2.8. Computational Resources

All investigations conducted in this paper were done using a laptop with a 6-core (12 threads) AMD Ryzen 5 Mobile 5500U processor with 16 GB of DDR4-2666 MHz memory. The codes that were executed on this hardware configuration were developed in Python programming language (Python version 3.9).

For balancing the original dataset using oversampling methods (random oversampling, ADASYN, SMOTE, and Borderline SMOTE) the imblearn library [45] (version 0.9) was used. To obtain symbolic expressions using GPSC the gplearn library [46] (version 0.4) was used. However, the random hyperparameter search method as well as the 5-fold cross-validation for the GPSC algorithm were developed from scratch. The evaluation metrics from scikit-learn [47] (version 1.13) were integrated into GPSC scripts with random hyperparameter search with 5-fold cross-validation and used each time the symbolic expression was obtained.

In multiclass case, the random hyperparameter search method for GPSC with 5-fold cross-validation was combined with OneVsRest Classifier (scikit-learn library function) to obtain symbolic expressions which could detect healthily, hepatitis C, fibrosis, and cirrhosis patients, respectively.

## 3. Results

In this section results of the conducted investigations are presented. Two types of investigations were considered i.e., the investigation of using GPSC to obtain the symbolic expression for detection of Hepatitis C patients and using GPSC to obtain the symbolic expression for detection of Hepatitis C disease progress (Hepatitis C, Fibrosis, and Cirrhosis). In the last subsection, the best symbolic expressions of each case are shown and the final evaluation of these expressions is performed on the original dataset.

### 3.1. The Symbolic Expression for Detection of Hepatitis C Patients

The Table 3 the list of randomly chosen hyperparameters with which highest values of classification metrics were achieved on each dataset variation are shown.

**Table 3.** The randomly chosen hyperparameters with which the symbolic expressions with highest classification accuracy were obtained.

| Dataset Type | GPSC Hyperparameters (Population_Size, Number_of_Generations, Tournament_Size, Initial_Depth, Crossover, Subtree_Muation, Hoist_Mutation, Point_Mutation, Stopping_Criteria, Max_Samples, Constant_Range, Parsimony_Coefficient) |
|---|---|
| Random oversampling | 463, 114, 34, (6, 12), 0.16, 0.13, 0.63, 0.069, $1 \times 10^{-5}$, 0.64, ($-8792.5$, 4309.56), $3.78 \times 10^{-5}$ |
| ADASYN | 493, 121, 46, (7, 12), 0.057, 0.39, 0.013, 0.53, $1 \times 10^{-5}$, 0.67, ($-4979.57$, 4518.54), $5.02 \times 10^{-5}$ |
| SMOTE | 428, 183, 25, (7, 12), 0.2, 0.41, 0.35, 0.029, 0.00084, 0.64, ($-7790.44$, 1461.71), $9.19 \times 10^{-6}$ |
| Borderline SMOTE | 384, 170, 34, (4, 11), 0.1, 0.26, 0.32, 0.31, $4 \times 10^{-6}$, 0.67, ($-4520.81$, 8562.5), $8.9 \times 10^{-5}$ |

As seen from Table 3 the population size in all four cases is near the upper bound i.e., near 500 while the number_of_generations and the tournament_size are near the lower bound. In all four cases, the hoist and point mutations (values in the range of 0.31–0.69) were dominating over crossover and subtree mutations (values in the range of 0.057–0.26). The stopping criteria values were very small in all cases (range $10^{-5}$–$10^{-6}$) however, the majority of GPSC executions were stopped due to the maximum number of generations value was reached not because fitness value dropped below the stopping criteria value. The max_samples was near the lower bound i.e., 0.6 (Table 2). Although the parsimony coefficient value was very low in all four cases (range $10^{-5}$–$10^{-6}$) the size of the population members did grow however this growth contributed to lowering the fitness measure value so no bloating phenomenon occurred. The graphical representation of the mean values of accuracy, precision, recall, AUC, and F1-score with standard deviation is shown in Figure 10.

Due to a small difference between mean and standard deviation values of evaluation metrics shown in Figure 10 all the results including the required average CPU time required to obtain these symbolic expressions are listed in Table 4.

From the obtained results shown in Figure 10, it can be seen that all the symbolic expressions have very high classification accuracy since mean *ACC*, *AUC*, Precision, Recall and F1-score values are all above 0.98. The highest evaluation metric values were achieved in the case of SMOTE, followed by ADASYN, BorderlineSMOTE, and Random Oversampling. To select the best symbolic expression for the binary classification problem the symbolic expression length was also measured. This length is measured by counting the number of elements (mathematical functions, and variables) inside the symbolic expressions. Based on that measure the longest symbolic expression was obtained in the ADASYN case followed by the Random Oversampling, SMOTE, and BorderlineSMOTE case. Since the symbolic expression obtained in the case of the BorderlineSMOTE dataset has slightly lower evaluation metric values than in the SMOTE case the best symbolic expression in the binary classification problem was chosen to be the symbolic expression obtained in the case of BorderlineSMOTE due to the small size.

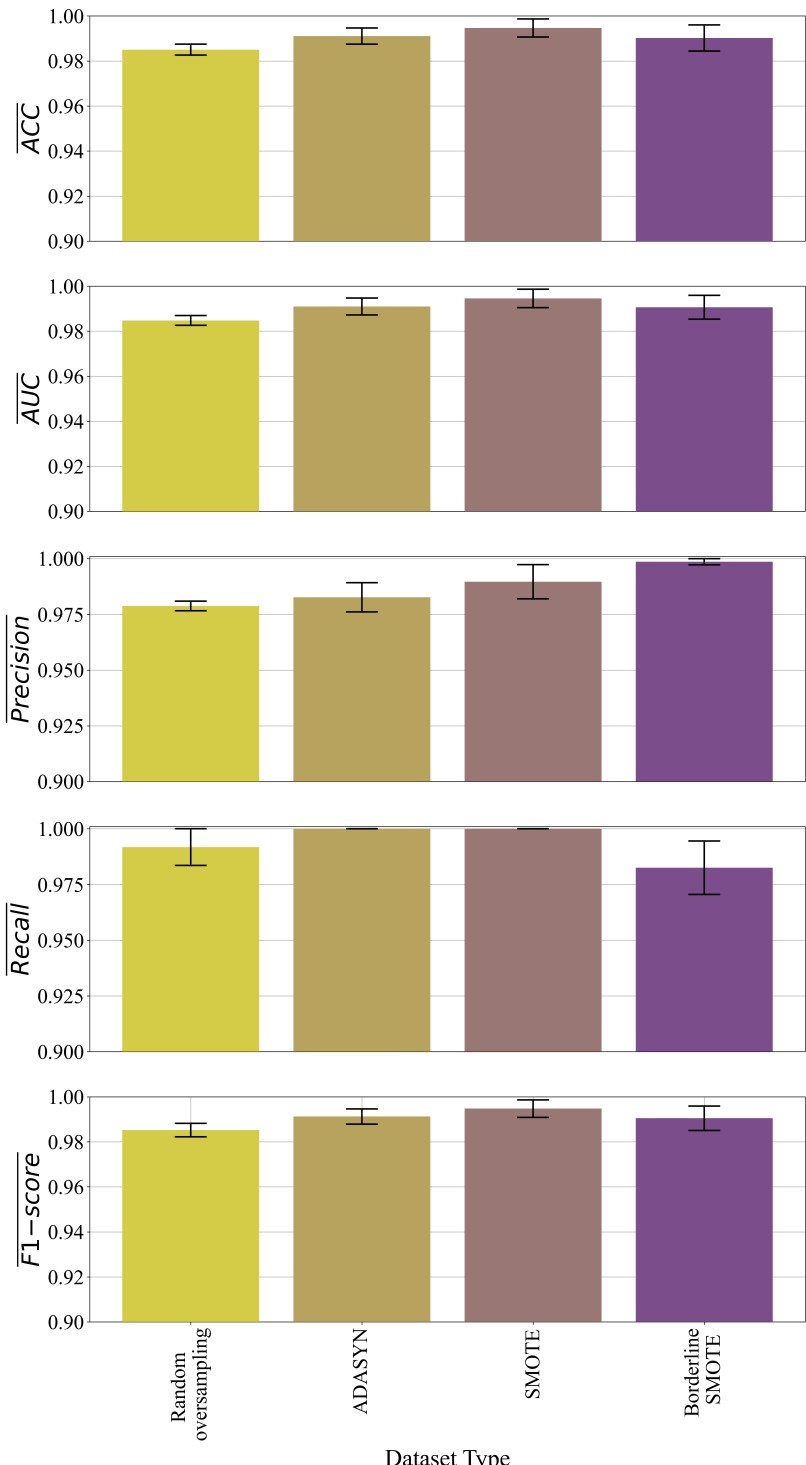

**Figure 10.** The mean and standard deviation (error bars) values of ACC, AUC, Precision, Recall for binary case.

**Table 4.** The numerical values of mean ACC, AUC, Precision, Recall, and F1-Score with standard deviation.

| Dataset Type | $\overline{ACC}$ $\pm SD(ACC)$ | $\overline{AUC}$ $\pm SD(AUC)$ | $\overline{Precision}$ $\pm SD(Precision)$ | $\overline{Recall}$ $\pm SD(Recall)$ | $\overline{F1-score}$ $\pm SD(F1-score)$ | Average CPU Execution Time min | Solution Length |
|---|---|---|---|---|---|---|---|
| Random Oversampling | 0.985 $\pm 2.46 \times 10^{-3}$ | 0.9847 $\pm 2.16 \times 10^{-3}$ | 0.978 $\pm 2.15 \times 10^{-3}$ | 0.991 $\pm 8.19 \times 10^{-3}$ | 0.985 $\pm 2.95 \times 10^{-3}$ | 120 | 91 |
| ADASYN | 0.991 $\pm 3.58 \times 10^{-3}$ | 0.99 $\pm 3.75 \times 10^{-3}$ | 0.982 $6.58 \times 10^{-3}$ | $1.0 \pm 0$ | 0.991 $\pm 3.3 \times 10^{-3}$ | 120 | 112 |
| SMOTE | 0.994 $\pm 4 \times 10^{-3}$ | 0.994 $\pm 4.11 \times 10^{-3}$ | 0.989 $\pm 7.68 \times 10^{-3}$ | $1.0 \pm 0$ | 0.994 $\pm 3.88 \times 10^{-3}$ | 120 | 423 |
| Borderline SMOTE | 0.99 $\pm 5.8 \times 10^{-3}$ | 0.99 $\pm 5.4 \times 10^{-3}$ | 0.998 $\pm 1.3 \times 10^{-3}$ | 0.98 $\pm 1.19 \times 10^{-3}$ | 0.99 $\pm 5.39 \times 10^{-3}$ | 120 | 60 |

The CPU time required for one GPSC execution with randomly selected hyperparameters and 5-fold cross-validation can be easily calculated. The process starts with 5-fold cross-validation on the training part of the dataset with randomly selected hyperparameters. The average CPU time for training and validation on each split in 5-fold cross-validation is 20 min. Since there are 5 splits in 5-fold cross-validation the total average CPU time is 100 min, If the 5-fold cross-validation passes the termination criteria i.e., the average values of evaluation metrics are higher than 0.97 then the final training/testing was performed. The training end of the final evaluation average CPU time is equal to 20 min. So the total average CPU execution time is equal to 120 min in all cases.

Generally, the major influence on the execution has the dataset size and the combination of used GPSC hyperparameters. Since the datasets in these investigations are very small (small number of samples) the combination of hyperparameters had a huge influence on GPSC execution time. The major problems regarding hyperparameters' influence on GPSC execution time can be the combination of large population size with a large number of generations, and the parsimony coefficient value. A large number of population members generally takes more time to process in each generation and if a large number of generations is set more time will be required to execute the GPSC algorithm. The parsimony coefficient value has a great influence on the evolution process and execution time of the GPSC algorithm. If the value is extremely small the size of the population members could grow rapidly which could cause extremely long execution times or execution failure. In this case, the values of the parsimony coefficient are very low which caused longer execution times even though the maximum number of generations was small.

*3.2. The Symbolic Expressions for Detection of Hepatitis C Stage*

The hyperparameters that were used to obtain the symbolic expression for each dataset variation with the highest classification accuracy are given in Table 5.

**Table 5.** The list of randomly chosen hyperparameters used to obtain best symbolic expressions for each dataset variation with high classification accuracy.

| Dataset Type | GPSC Hyperparameters (Population_size, Number_of_Generations, Tournament_Size, Initial_Depth, Crossover, Subtree_Muation, Hoist_Mutation, Point_Mutation, Stopping_Criteria, Max_Samples, Constant_Range, Parsimony_Coefficient) |
|---|---|
| Random oversampling | 297, 95, 85, (6, 8), 0.53, 0.024, 0.186, 0.256, $8 \times 10^{-5}$, 0.99, $(-7110.15, 9285.3)$, $1.63 \times 10^{-6}$ |
| ADASYN | 397, 148, 122, (5, 8), 0.53, 0.26, 0.113, 0.085, $5 \times 10^{-5}$, 0.61, $(-2905.8, 464.71)$, $1.17 \times 10^{-5}$ |
| SMOTE | 422, 102, 21, (7, 12), 0.45, 0.13, 0.25, 0.16, $8 \times 10^{-5}$, 0.67, $(-5675.45, 5426.23)$, $6.76 \times 10^{-6}$ |
| Borderline SMOTE | 541, 119, 14, (7, 8), 0.5, 0.13, 0.19, 0.16, $8 \times 10^{-5}$ 0.63, $(-3670.2, 4287.4)$, $6.76 \times 10^{-6}$ |

As seen form Table 5 the crossover coefficient is the dominating genetic operation when compared to other three genetic operations. The maximum number of samples used from training dataset was set to 0.99 in case of random oversampled dataset and set to around 0.6 for remaining three cases. The parsimony coefficient was pretty low in all cases however, bloat phenomenon did not occur. Each GPSC execution was terminated after maximum number of generations was reached which means that in none of GPSC executions non of the population members reached the predefined lowest value of the fitness function.

The mean values of *ACC*, *AUC*, *Precision*, *Recall*, and *F1-score* with standard deviation is shown in Figure 11 while numerical results are listed in Table 6.

From Figure 11 and Table 6 it can be seen that the best classification accuracy was achieved in the case of the dataset balanced with random oversampling method followed by the dataset balanced with Borderline SMOTE, ADASYN, and SMOTE method. In terms of symbolic expressions length, the largest symbolic expressions were obtained in the case of random oversampling dataset followed by ADASYN, SMOTE, and BorderlineSMOTE. Based on the evaluation metric values and the length of symbolic expressions the best symbolic expressions were obtained in the case of BorderlineSMOTE since the size of these symbolic expressions is the smallest and evaluation metric values are near those obtained with the random oversampling dataset. All four symbolic expressions are shown in the following subsection.

The average CPU execution time in all four cases is the same i.e., 480 min. The longer execution time can be attributed to utilization of OneVsRestClassifier in combination with 5-fold cross-validation. This means that for each class 5-fold cross-valdiation is performed i.e., the total number of GPSC executions is 20. Each GP execution on average lasts for 20 min so that is total of 400 min. When this process is complete the evaluation metrics are averaged and if the termination criteria is passed final train/test is performed with same hyperparameters and OneVsRestClassifier. So final training has to be executed 4 times due to 4 different cases and each GPSC execution lasts for 20 min so the final training lasts for additional 80 min. In total average GPSC execution time is 480 min.

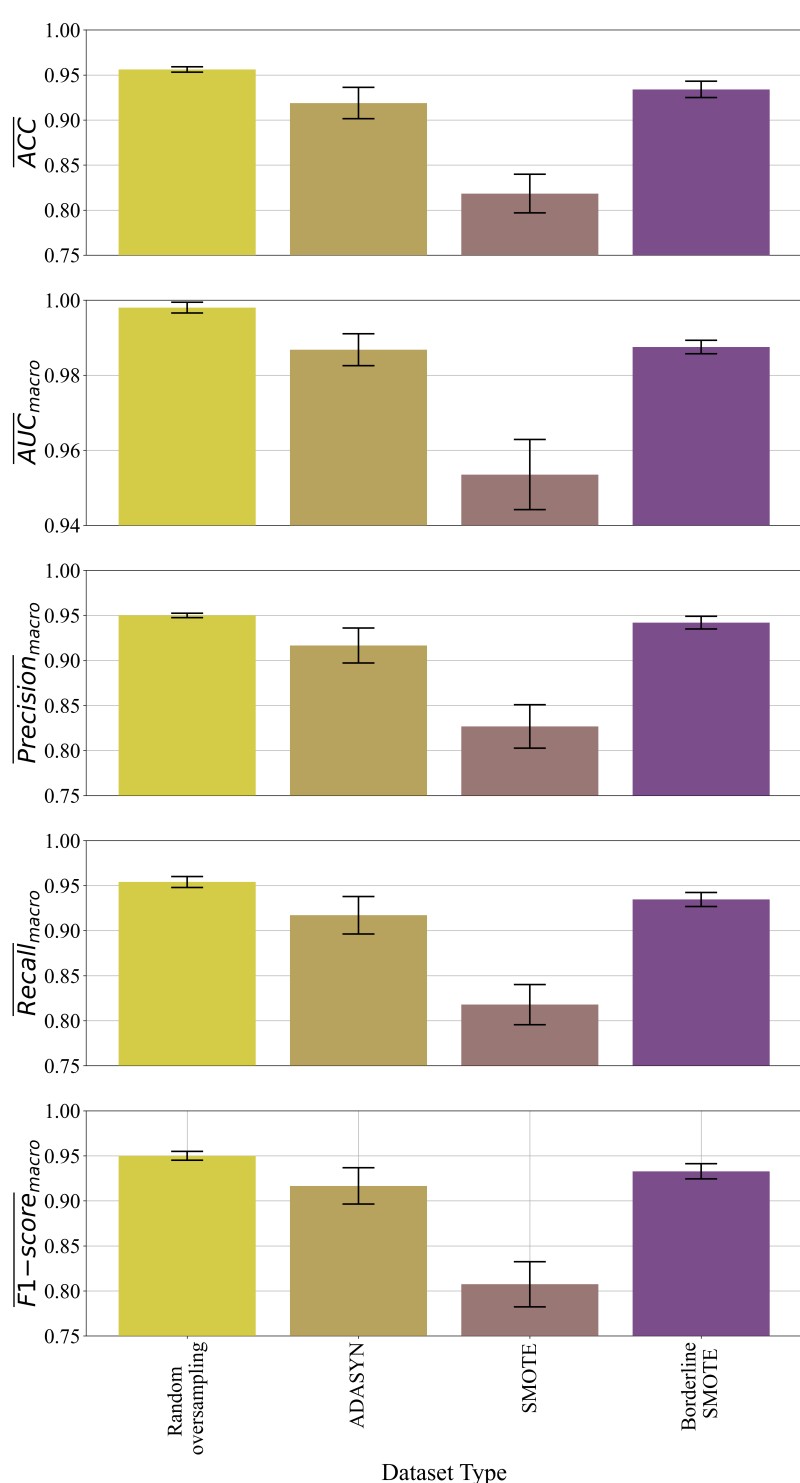

**Figure 11.** The graphical representation of mean values of ACC, AUC, precision, recall, F1-Score with standard deviation.

**Table 6.** The numerical values of mean ACC, AUC, Precision, Recall, and F1-Score with standard deviation.

| Data Type | $\overline{ACC}$ $\pm SD(ACC)$ | $\overline{AUC}$ $\pm SD(AUC)$ | $\overline{Precision}$ $\pm SD(Precision)$ | $\overline{Recall}$ $\pm SD(Recall)$ | $\overline{F1-score}$ $\pm SD(F1-score)$ | Average CPU Execution Time min | Symbolic Expression Length (1/2/3/4) |
|---|---|---|---|---|---|---|---|
| Random Oversampling | 0.956 $\pm 2.99 \times 10^{-3}$ | 0.998 $\pm 1.42 \times 10^{-4}$ | 0.95 $\pm 2.5 \times 10^{-3}$ | 0.954 $\pm 6 \times 10^{-3}$ | 0.95 $\pm 4.9 \times 10^{-3}$ | 480 | 294/1961/998/2882 |
| ADASYN | 0.918 $\pm 1.74 \times 10^{-2}$ | 0.986 $\pm 4.25 \times 10^{-3}$ | 0.916 $\pm 1.93 \times 10^{-2}$ | 0.917 $\pm 2 \times 10^{-2}$ | 0.916 $\pm 2 \times 10^{-2}$ | 480 | 696/90/675/216 |
| SMOTE | 0.81 $\pm 2.1 \times 10^{-2}$ | 0.95 $\pm 9.3 \times 10^{-3}$ | 0.82 $\pm 2.41 \times 10^{-2}$ | 0.81 $\pm 2.2 \times 10^{-3}$ | 0.807 $\pm 2.5 \times 10^{-2}$ | 480 | 172/19/882/558 |
| Borderline SMOTE | 0.934 $\pm 9 \times 10^{-3}$ | 0.987 $\pm 1.8 \times 10^{-3}$ | 0.942 $\pm 6.9 \times 10^{-3}$ | 0.934 $\pm 7.84 \times 10^{-3}$ | 0.932 $\pm 8.4 \times 10^{-3}$ | 480 | 417/148/471/118 |

### 3.3. Best Symbolic Expressions and Final Evaluation

Based on the conducted investigation and obtained results it can be concluded that the best symbolic expression in binary and multiclass problem are those symbolic expressions that were obtained on datasets balanced with BorderlineSMOTE method. The symbolic expression for binary problem (binary classification) can be written in the following form:

$$
\begin{aligned}
y_1 \;=\; & \left| 2098.97 \right| \Bigg( X_9 (\min(\frac{X_5 + X_6}{\min(\sqrt{X_{11}}X_5, X_3)}, X_9) - \min((X_4 \\
- \; & X_9(\min(\frac{X_5 + \tan(X_6)}{\min(\sqrt{X_{11}}X_5, X_3)}, X_0) - \min((X_4 - \log_2(X_3)) - X_1, X_3 \frac{\frac{X_4}{X_6}}{X_{10}}))) \\
- \; & X_1, X_3 \frac{\frac{X_4}{X_6}}{X_{10}})) \Bigg).
\end{aligned}
\tag{12}
$$

As seen from Equation (12) the best symbolic expression obtained in case of dataset balanced with BorderlineSMOTE method the input variables that end up in the symbolic expressions are $X_0$, $X_1$, $X_3$, $X_4$, $X_5$, $X_6$, $X_9$, $X_{10}$, and $X_{11}$. From Table 1 it can be seen that these variables are age, sex, ALP, ALT, AST, BIL, CREA, GGT, and PROT.

In case of multiclass problem the best case in terms of symbolic expression accuracy was achieved in case of Random Oversampling, and BorderlineSMOTE. However, the random oversampling dataset contains the same points as the original dataset and they are oversampled while the Borderline SMOTE contains samples around the original points so symbolic expression obtained for this dataset are better in terms of generalization and robustness although classification performance is lower than in random oversampling case. The equations obtained in case of Borderline SMOTE method consist of 4 equations and these are:

- the equation for detection of healthy patients ($y_{21}$),
- the equation for detection of Hepatitis C patients ($y_{22}$),
- the equation for detection of Fibrosis patients ($y_{23}$), and
- the equation for detection of Cirrhosis patients ($y_{24}$).

The equations can be written as

$$
\begin{aligned}
y_{21} &= \min\Bigg(\log_2(X_{10})\Bigg(\min(12.34, X_5 - \cos(\sqrt[3]{\sqrt{\sin(\sin(\min(|\sqrt[3]{X_3}|, \min(\sqrt[3]{X_5}\sqrt[3]{\max(\cos(\sqrt[3]{X_3}), \sqrt[3]{X_5})}, X_5)))))}}\\
&\quad \sin(\min(|\log(X_2)|, \min(|\tfrac{|X_9|}{2.18}|, \max(\tfrac{X_4}{X_{11}}, \sqrt[3]{X_5}) - \sin(\min(\log(X_2), \min(|\log(X_5) - \max(\log_{10}(\tfrac{X_2}{X_4}\log(X_2)),\\
&\quad \max\Bigg(|\Big(\max(\min(\log(X_2), \min(|\tfrac{(X_3 + \frac{|X_9|}{2.18}) - \sqrt[3]{\sin(\cos(\sqrt[3]{\sqrt{X_8}}))}}{|\log(X_2)|}|, \max(\tfrac{X_4}{X_{11}}, \sqrt[3]{X_5}) - \min(\sqrt{X_0}, \log_2(X_{10}))\\
&\quad \sin(\min(\log(X_2), \min(|\log(X_5) - \max(\log_{10}(\tfrac{X_2}{X_4}\log(X_2)), \max(\log_{10}(\cos(X_6)), \max(\tfrac{X_9}{X_2}, \sqrt[3]{X_5})))|,\\
&\quad \log_2(\sqrt[3]{\log_{10}(X_4)}))))), \sqrt[3]{X_5}) - \cos\Big(\big(\log_{10}(\min(1.09, (\cos(|\sqrt[3]{X_0}|)\max(\cos(\tfrac{|\frac{\log(X_5)}{\min(|\sqrt[3]{X_3}|, \sqrt{\min(X_6, X_2)})}|}{X_8}), \sqrt[3]{X_5}))\\
&\quad \sin(|3214.39|))))^{\frac{1}{3}}\Big)\Big) \Big/ \sqrt[3]{\tfrac{X_6}{X_2}}|, \sqrt[3]{\tfrac{X_6}{X_2}}\Big)|, \log_2(\sqrt[3]{\log_{10}(X_4)}))))\min(\sqrt{X_0}, \max(\cos(\log_2(X_{10})),\\
&\quad \max(|\big(\max(\tfrac{X_9}{X_2}, \sqrt[3]{X_5}) - \cos\big(\big(\cos(|\log(X_{11})\big/\big(\big(\max(\min(\log(X_2), \min(|\tfrac{|\log_2(|-3455.66|)|}{2.18}|,\\
&\quad \max(\tfrac{X_4}{X_{11}}, \sqrt[3]{X_5}) - \min(\sqrt{X_0}, \max(\cos(\log_2(X_{10})), \max(|\big(\max(\tfrac{X_9}{X_2}, \sqrt[3]{X_5}) - \cos\big(\big(\log_{10}(\max(\sqrt[3]{\log_2(X_{10})},\\
&\quad \max(\sqrt[3]{\tfrac{\max(\cos(\sqrt[3]{X_3}), \sqrt[3]{X_5}) - \cos(\log_2(X_{10}))}{\sqrt[3]{X_5}}}, \max(\tfrac{X_9}{X_5}, \sqrt[3]{X_5}))) - \sqrt[3]{\cos(X_7)\sqrt{1601.1}}\big)^{\frac{1}{3}}\big)\big)\Big/\\
&\quad \log(\log(\log_2(\cos(X_7 X_1))))|, \max(\tfrac{X_9}{X_2}, \sqrt[3]{X_5}))) - \big(\min(\cos(|\tfrac{\log(X_5)}{\min(|\sqrt[3]{X_0}|, \sqrt{\log(\log_{10}(X_{10} + X_5))})}|),\\
&\quad \max(-\tfrac{X_9}{2236.3}, \sqrt[3]{X_0}) - \sin(\tfrac{X_9}{X_2})))^{\frac{1}{9}}(X_7 - X_1))), \sqrt[3]{X_5}) - \cos\big(\big(\min(|\sqrt[3]{X_3}|,\\
&\quad \big(\min(|\tfrac{\max(\tfrac{X_9}{X_2}, \sqrt[3]{X_5}) - \cos(\sqrt[3]{\log_{10}(\min(\sqrt[3]{\log_2(\sqrt[3]{\log_{10}(X_4)})}, (\cos(|\sqrt[3]{X_0}|)\sqrt{X_9}\sin(|3214.39|)))})}}{\sqrt[3]{\tfrac{X_6}{X_2}}}|,\\
&\quad X_2)\tfrac{1}{2})^{\frac{1}{3}}\big)\big/\sqrt[3]{\tfrac{X_6}{X_2}}|))^{\frac{1}{3}}\big)\big)\big/\sqrt[3]{\tfrac{X_6}{X_2}}|, \max(\tfrac{X_9}{X_2}, \sqrt[3]{X_5})))\\
&\quad - (\min(\cos(|\tfrac{\log(X_5)}{\min(|\log(\log(\log(\log_2(\cos(X_7 X_1))))))|, \sqrt{\min(X_6, X_2)})}|), X_9))^{\frac{1}{9}})))\Bigg)^{\frac{1}{3}}, X_5\Bigg),
\end{aligned}
\tag{13}
$$

$$
\begin{aligned}
y_{22} &= \log_2\Bigg(\log_{10}\Bigg(\log_2\Bigg(\sin(\log_2(\sin(\sin(\min(\log(|X_5 - X_5|)\big/(|\big(\log_2(\max(\log_2(\log_2(\log_2(\log(X_0\\
&\quad + \log_2(\log_{10}(|\sqrt{\tan(\log_{10}(X_7))}|)))) + \sin(\min(0, \log_{10}(\tan(\log_{10}(|\sin(\min(\sqrt[3]{\sin(\log(X_0))}\\
&\quad - \sqrt[3]{\cos(\max(X_9, X_{11}))}, \max(|\max(\log_2(X_{10}), \sqrt{X_1})|, \tfrac{\max(X_{10} + X_{11}, \cos(X_4))}{\log(\log_2(X_1))})))|))))))))),\\
&\quad \sin(\tan(\log_{10}(\log_{10}(X_7 - X_0)\log(X_8)))))))^{\frac{1}{2}}|), \sin(\tan(\log_{10}(\tfrac{X_7}{X_3}))))))) + \log_2(\log_2(\log_{10}(|\log_{10}(X_1)|))\\
&\quad \sqrt[3]{(\sin(\log_2(\log_{10}(|\log_{10}(X_1)|))) + \log_{10}(\log_{10}(\tfrac{X_7}{X_0})\log_{10}(\tfrac{X_7}{X_3})) + X_5))) + \log_{10}(\log(X_8)\log_{10}(\tfrac{X_7}{X_0})))^{\frac{1}{2}}}\Bigg)\Bigg)\Bigg),
\end{aligned}
\tag{14}
$$

$$
\begin{aligned}
y_{23} =\ & \log_2(|\cos(\log_2(\cos(\cos(\log_2(\log_2(X_{11}))))\cos(\log_2(\cos(\log_2(\min(X_{11},\cos(\log_2(\min(X_{11},\log_{10}(X_5)))) \\
& \tan(\log_2(\log_2(X_0)))))))\tan(\log_2(\log_2(X_0))))))\tan(\cos(\log_2(\log_2(X_0)))) \\
& \cos(\log_2(\cos(\log_2(\cos(\cos(\log_2(\log_2(X_0)))))))) \\
& \tan(\cos(\log_2(\log_2(\min(X_{11},\log(\min(X_3,X_6))))))) \cos(\cos(\log_2(\cos(\log_2(\log_2(\min(X_{10},9.695\Big/ \\
& \Big(\Big(|\cos(\log_2(\tan(\cos(X_{11}-X_3)\log(\cos(\sqrt[3]{\cos(\min(X_0,X_2))})\cos(\sqrt{X_8})))\tan(\cos(\log(\log(\log_2(X_{11}-X_3)))) \\
& \cos((\log_2(\cos(\log_2(\cos(\log_2(\log_2(\min(X_{11},\log(\min(X_3,X_6))))))))\cos(\cos(\log_2(\cos(\log_2(\log_2(\min(X_{10}, \\
& \tfrac{9.695}{\frac{\log(\min(X_3,\log_{10}(X_5)))}{|X_4|}})))))\tfrac{X_2}{X_9}))\tfrac{X_2}{X_9})))\tan(\cos(\log(\log(\min(X_3,\log_{10}(X_5))))))\cos(\cos(\log_2(\log_2(X_{11}))) \\
& \cos(\log_2(\cos(\log_2(\min(X_{11},\log_{10}(X_5))))\tan(\log_2(\log_2(X_0)))))))))\tan(\log_2(\log_2(X_0)))\tan(\tfrac{X_2}{X_9}))))) \\
& \tan(\cos(\log_2(\log_2(\min(X_{10},9.695\Big/\Big(\Big(|\log_2(\log(X_7))|\cos(\cos(\log_2(\log_2(X_0)))) \\
& \tan(\log_2(\cos(\log(\log(\min(X_3,\log_{10}(X_5))))))\log_2(\min(X_{11},\log_{10}(X_5)))) \\
& \tan(\cos(\log_2(\cos(\log_2(\log_2(\cos(\log_2(\log_2(X_9-X_3)))\tan(\log_2(\log_2(X_0)))))) \\
& \tan(\cos(\log_2(\log_2(\cos(\tan(\log_2(\log_2(X_0))))\tan(\tfrac{X_2}{X_9}))))\cos((\cos(\log_2(\log_2(\cos(\log_2(\log(|X_6|))) \\
& \cos(\cos(\log_2(\log_2(X_{11}-X_3)))\tan(\log_2(\log_2(X_0)))))))\tan(\log_2(\log_2(X_0)))\tan(\tfrac{X_2}{X_9})))))\log_{10}(X_5)))|\Big) \\
& \Big/|X_4|))))))|\Big)\Big/|X_4|))))))\tan(\log_2(\log_2(X_0)))\tfrac{X_2}{X_9})))))))|\cos(\log_2(\log_2(\cos(\log_2(\log_2(X_{11}-X_3))) \\
& \tan(\log_2(\log_2(X_0)))))))\tan(\cos(\log_2(X_7))|\cos(\log_2(\cos(\log_2(\log_2(\cos(\log_2(X_2)) \\
& \tan(\tfrac{X_2}{X_9}))))\tan(\cos(\log_2(\cos(\log_2(\cos(\log_2(\log_2(\min(X_{11},\log(\min(X_3,X_6)))))) \\
& \cos(\cos(\log_2(\cos(\log_2(\log_2(\min(X_{10},\tfrac{9.695}{\frac{\log(\min(X_3,\log_{10}(X_5)))}{|X_4|}}))))) \\
& \tan(\log_2(\log_2(X_0)))))\tfrac{X_2}{X_9})))\tan(\cos(\log(\log(\min(X_3,\log_{10}(X_5)))))) \\
& \cos(\cos(\log_2(\log_2(X_{11})))\cos(\log_2(\cos(\log_2(\min(X_{11},\log_{10}(X_5)))) \\
& \tan(\log_2(\log_2(X_0))))))))))\cos(\cos(\log_2(\cos(\log_2(\log_2(X_{11}-X_3))))) \\
& \tan(\tfrac{X_2}{X_9})))))|\cos(\log_2(\log_2(\cos(\log_2(\log_2(X_9-X_3)))\tan(\log_2(\log_2(X_0)))))) \\
& \tan(\cos(\log_2(\log_2(\cos(\tan(\log_2(\log_2(X_0))))\tan(\tfrac{X_2}{X_9}))))\tan(\cos(\log_2(\log_2(\min(X_{11},\log_{10}(X_5))))\sqrt{\tfrac{X_2}{X_9}}))|)|)|),
\end{aligned} \tag{15}
$$

$$
\begin{aligned}
y_{24} =\ & \log_2(\max(X_7,X_5))-\max(\max((\log_2(X_9)-(X_4X_7)(\log_2(\min(X_3,X_5)) \\
- & \max(\min(X_5,X_4)\log_2(\log_{10}(\max(X_1,\log_2(2.36-\tfrac{X_5}{\log_2(\min(X_3,X_5))}) \\
- & (X_7X_9)(\log_2(\min(X_0,2.36))-\max((\min(X_0,2.36) \\
- & \max(\min(\tan(\log_2(\min(X_7X_9,\cos(X_0))+\sqrt[3]{\log_2(X_8)})) \\
& ,\log_2(\log_{10}(\max(X_1,\log_2(\log_{10}(\log_2(X_9)))))))((\cos(X_2)+(X_6+9.662)) \\
+ & \log_{10}(X_7X_9))+\sqrt[3]{X_9}),\min(X_7,X_5)))-X_2,\min(X_7,X_5))))))-X_2,\min(X_7,X_5)))) \\
- & \tfrac{X_5}{\log_2(X_7X_9)},X_7)-X_9,\min(X_7,X_4)).
\end{aligned} \tag{16}
$$

In the previous set of symbolic expressions all input variables $(X_1,\ldots,X_{11})$ are included. The final evaluation of the previous system of symbolic expressions on original dataset is shown in the following subsection.

Final Evaluation

Since the original dataset was not used in previous investigations it will be used here to perform final evaluation of the best previously presented symbolic expressions. However, in the binary case the dataset samples labeled with classes 1, 2, 3 are all put together under one class 1. So the entire dataset in binary classification has two class 0—Healthy and 1—Hepatitis C. The procedure of evaluating the symbolic expressions are as follows:

- use input values of the original dataset in symbolic expressions to compute the output,
- use the calculated output as input in Sigmoid function as decision function to obtain the class output (0 or 1),
- compare the output from decision function with the real output to compute ACC, AUC, Precision, Recall, and F1-Score.

The evaluation metric values obtained with symbolic expression generated using dataset balanced with Borderline SMOTE method applied on the original dataset are listed in Table 7.

**Table 7.** The results of evaluation metric values obtained with application of the best symbolic expression ($y_1$) in binary problem on the original dataset.

| Evaluation Metric | Values |
|:---:|:---:|
| *ACC* | 0.9932 |
| *AUC* | 0.9722 |
| *Precision* | 0.98148 |
| *Recall* | 0.9464 |
| *F1 − Score* | 0.9636 |

As seen from Table 7 the results of the best symbolic expression applied on the original dataset are slightly lower than the results obtained on dataset balanced with Borderline SMOTE method i.e., the dataset used to obtain the best symbolic expression in binary problem. The macro averaging evaluation metric values in multi class case is shown in Table 8.

**Table 8.** The results of evaluation metric values obtained with application of the best symbolic expressions ($y_{21}, y_{22}, y_{23}$, and $y_{24}$) in multiclass on the original dataset.

| Evaluation Metrics | Values |
|:---:|:---:|
| *ACC* | 0.983 |
| *AUC* | 0.85 |
| *Precision* | 0.74 |
| *Recall* | 0.72 |
| *F1 − Score* | 0.721 |

The results for a multiclass problem showed that the obtained evaluation metric values are lower than those obtained in case of BorderlineSMOTE dataset (on which symbolic expressions were obtained). This can be attributed to high imbalance of the original dataset and very small number of fibrosis and cirrhosis class samples.

## 4. Discussion

The conducted investigation showed the procedure of how highly imbalanced datasets can be balanced and used to obtain the symbolic expression for the detection of hepatitis C patients from blood samples. The kernel PCA method provided a better insight into the distribution of class samples in the kernel PCA plane. The random oversampling method generated enlarged the number of samples however those are the same samples as the originals. Among the remaining three balancing methods the distribution of class samples in the kernel PCA plane showed that class samples overlap which could be a problem for the ML algorithm to distinguish between the overlapping classes. The ADASN and SMOTE

method enlarged the area of class samples and overlapping between class samples is the most evident in those two cases. However, in the case of the Borderline SMOTE balancing method the distribution of class samples is more condensed i.e., the synthetic samples are generated around the original class samples.

In binary problem, the best symbolic expression was obtained in the case of the dataset balanced with the BorderlineSMOTE method as seen from Figure 10 and Table 4. However, the symbolic expression obtained in the case of the dataset balanced with SMOTE method showed similar classification accuracy but the size of the symbolic expression, in this case, is much larger than the symbolic expression obtained in the Borderline SMOTE case. Based on size and evaluation metric values the best symbolic expression in binary problem was the symbolic expression obtained on dataset balanced with BorderlineS-MOTE method. The symbolic expression in the final evaluation on the original dataset showed similar classification accuracy (Table 7) as in the case of the Borderline SMOTE method. Regarding the GPSC hyperparameters in binary problems, the hoist and point mutation were dominating genetic operations. Due to the low parsimony coefficient value ($9.19 \times 10^{-6}$) in the SMOTE case the large symbolic expression was obtained however the bloat phenomenon did not occur because the GPSC execution time was similar to other cases and classification accuracy is similar to one obtained in BorderlineSMOTE case.

In multi-class problems, the best symbolic expressions were obtained in the case of dataset oversampled with random oversampling and the Borderline SMOTE method. However, in the case of SMOTE and ADAYN datasets, the obtained symbolic expressions performed poorly i.e., the results of evaluation metrics are in the range from 0.8–0.99. The symbolic expressions obtained in the case of the BorderlineSMOTE method are smaller when compared to those symbolic expressions obtained in the case of the random over-sampling method. The classification accuracy of the symbolic expressions obtained in the case of Borderline SMOTE is slightly lower than of those symbolic expressions obtained with random oversampling. Regarding the size the smallest size of symbolic expression was obtained in the BorderlineSMOTE case while the largest was in the case of Random Oversampling. Due to the smallest symbolic expressions and the pretty high evaluation metric values, the symbolic expressions in the multiclass problem were those expressions obtained with a dataset balanced with the BorderlineSMOTE method.

The final evaluation performed on the original dataset showed the poor performance of obtained symbolic expressions when compared to the results achieved on the dataset balanced with Borderline SMOTE Method. In the multiclass problem, the crossover was the dominating genetic operation i.e., its value in all four cases was above 0.45 (Table 5). The parsimony coefficient is the lowest in the case of the random oversampling method so this could be a reason why the symbolic expressions are so large and the classification accuracy is insignificantly better than the symbolic expressions obtained in the case of dataset balanced BorderlineSMOTE method. The final evaluation of the best symbolic expressions applied on the original dataset showed poor performance of these symbolic expressions (Table 8). The low performance of these symbolic expressions can be attributed to a high imbalance of the original dataset (the high number of healthy patients versus the small number of Hepatitis C patients and patients with fibrosis and cirrhosis).

## 5. Conclusions

In this paper, the GPSC was used with a random hyperparameter search method and 5-fold cross-validation to obtain symbolic expressions for the detection of hepatitis C patients as well as determining the hepatitis C stage (Hepatitis C, Fibrosis, and Cirrhosis) using a dataset containing blood samples. However, the original dataset was highly imbalanced so before using GPSC the dataset balancing techniques had to be applied such as random oversampling, SMOTE, ADASYN, and BorderlineSMOTE. Unfortunately due to a large class imbalance in both binary and especially the multiclass case of the original dataset was not used for generating symbolic expressions using the GPSC method. The original dataset was used to perform final tests on the best symbolic expressions

obtained on datasets balanced with other methods. Based on the conducted investigations, the following conclusions are:

- using GPSC with random hyperparameters search and 5-fold cross-validation the symbolic expression can be obtained which can detect the Hepatitis-C patients with high classification accuracy,
- using GPSC with One Versus Rest Classifier, random hyper-parameter search method and 5-fold cross-validation it is possible to obtain the symbolic expressions that can detect the hepatitis C patients and their disease progression (Hepatitis-C, Fibrosis and Cirrhosis). However, required average CPU execution time is quite long but this is relative since the execution times mostly depends on the used computational resources,
- the application of different balancing methods can synthetically balanced the class samples which in the end can improve the classification accuracy of obtained symbolic expressions.

This investigation showed how imbalanced dataset can be balanced using over-sampling methods and used in GPSC algorithm with which symbolic expression can be obtained. To achieve higher classification accurracies in terms of hepatitis C detection and the stage of the disease the random hyperparameter search method for GPSC with 5-fold cross-validation was employed. The used method also addresses the issue of the so-called black-box models, by providing a possibility of interpreting the generated classification models.

In future investigations, the original dataset will be enlarged if possible. The idea is to achieve an equal number of samples through all the dataset classes. The enlarged dataset could be a good starting point to develop a system of symbolic expressions for detecting Hepatitis C patients and the possible disease progression. Further investigation and tuning of GPSC hyperparameters are required to obtain very small symbolic expressions that have high classification accuracy. One of the limitations of this approach is the parsimony coefficient which has a large impact on the performance of GPSC. The influence of this hyperparameter will be reduced by ensuring that the dataset used is balanced and that correlation between variables is reasonably high.

**Author Contributions:** Conceptualization, N.A. and Z.C.; methodology, N.A., Z.C and I.L.; software, N.A. and S.B.Š.; validation, N.A., I.L. and S.B.Š.; formal analysis, N.A., I.L. and S.B.Š.; investigation, N.A., I.L. and S.B.Š.; resources, N.A., Z.C. and S.B.Š.; data curation, I.L. and S.B.Š.; writing—original draft preparation, N.A., I.L. and S.B.Š.; writing—review and editing, N.A., I.L. and S.B.Š.; visualization, I.L. and S.B.Š.; supervision, Z.C.; project administration, Z.C.; funding acquisition, Z.C. All authors have read and agreed to the published version of the manuscript.

**Funding:** This research received no external funding.

**Institutional Review Board Statement:** Not applicable.

**Informed Consent Statement:** Not applicable.

**Data Availability Statement:** Not applicable.

**Acknowledgments:** This research has been (partly) supported by the CEEPUS network CIII-HR-0108, European Regional Development Fund under the grant KK.01.1.1.01.0009 (DATACROSS), project CEKOM under the grant KK.01.2.2.03.0004, Erasmus+ project WICT under the grant 2021-1-HR01-KA220-HED-000031177, and University of Rijeka scientific grant uniri-tehnic-18-275-1447.

**Conflicts of Interest:** The authors declare no conflict of interest.

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
