# Peer review of "The Development of Symbolic Expressions for the Detection of Hepatitis C Patients and the Disease Progression from Blood Parameters Using Genetic Programming-Symbolic Classification Algorithm"

_applsci, doi:10.3390/app13010574_

Round 1
Reviewer 1 Report
1. Good work. Your work also improves the much needed interpretability of the classification. You may want to mention this advantage as well - it seems to be currently missing in the paper.
2. From the PCA plots and the symbolic equations, the data is clearly non-linear. Using t-SNE or other non-linear techniques instead of PCA (which is linear) seems a better approach.
3. It may be a better idea to try out the Machine Learning models used in the current literature like KNN, SVM, and ensemble methods like GBM using the same oversampling techniques and use those results as baseline for comparison. It will give the readers more confidence in your work.
4. There is similar work for a related problem (liver cancer) that can probably be cited to show that the idea has more backing.
5. Formula for F1 score in equation (12) is incorrect. Please check.
6. Future directions can be strengthened and limitations identified.
7. Requires better proof-reading to correct language errors like “is two show,” “However The main …”, “poorly imbalanced dataset” etc.
Author Response
The authors want to thank the reviewer for his time and effort to give constructive comments and suggestions which could greatly improve the manuscript's quality. The authors of this manuscript hope that the changes made in this manuscript will provide a suitable scientific contribution.
- Good work. Your work also improves the much-needed interpretability of the classification. You may want to mention this advantage as well - it seems to be currently missing in the paper.
Answer: The authors want to thank the reviewer for reading the manuscript and acknowledging its quality. The comment regarding the much-needed interpretability of the classification done in this paper is addressed in the Conclusion section. Citing from the revised version of the manuscript (Conclusion section): “The used method also addresses the issue of the so-called black-box models, by providing a possibility of interpreting the generated classification models.”
- From the PCA plots and the symbolic equations, the data is clearly non-linear. Using t-SNE or other non-linear techniques instead of PCA (which is linear) seems a better approach.
Answer: The idea was to somehow graphically represent class samples to prove that the dataset was balanced using different balancing methods and to see where are these synthetic samples created. The authors agree with the comment made by the reviewer and the classic PCA (linear) was omitted from the manuscript. Initially, we considered the suggested t-SNE method, however, we have decided to use Kernel PCA which is also a non-linear dimensionality reduction technique and is good for the graphical representation of class samples.
- It may be a better idea to try out the Machine Learning models used in the current literature like KNN, SVM, and ensemble methods like GBM using the same oversampling techniques and use those results as baseline for comparison. It will give the readers more confidence in your work.
Answer: Using KNN, SVM, and ensemble methods would generally produce trained algorithms whit high classification accuracy. However, this was not the idea of this paper. As stated in the Introduction of this paper the idea was to investigate if using the GPSC algorithm a symbolic expression could be generated which could be used to detect Hepatitis C patients and disease progression from blood samples. Mathematical equations are easier to understand and use by the general public than trained AI/ML models. They acquire lower computational resources (less memory, and computational power) to produce an output when compared to other trained AI/ML models.
- There is similar work for a related problem (liver cancer) that can probably be cited to show that the idea has more backing.
Answer: The authors have searched the literature regarding liver cancer and the literature was included in the Introduction section just after the 106 lines in the original/revised version of the manuscript. Citing a newly created paragraph from a revised version of the manuscript (lines 107-124):
“There is some notable research in which AI and ML algorithms have been used in the detection of liver cancer. In [15] the authors have used the support vector machines method for identifying the liver cancer tumor for ultrasound images. The results showed that using this method a classification accuracy of 96.72% was achieved. The ANN and logistic regression have been used in [16] to develop a model for predicting the development of liver cancer within 6 years of diagnosis with type II diabetes. The best results were achieved with ANN in terms of sensitivity (75.7%), specificity (75.5%), and the area under the receiver operating characteristic curve (87.3%). The ANN and classification of regression tree have been used in [17] on a dataset collected from the cancer registration database in Northern Taiwan medical center from 2004 to 2008 to predict the survival of patients with liver cancer. The best results were achieved with ANN in terms of accuracy (87%), sensitivity (88%), specificity (87%), and area under the receiver operating characteristic curve (91.5\%). The ensemble method has been developed and used in \cite{muflikhah2020prediction} to predict liver cancer in patients based on DNA sequence. Initially, the Naive Bayes, (GLM), kNN, SVM and C5.0 Decision Tree have been considered as elements of the ensemble method however, the best results were achieved with the ensemble method consisting of C5.0 Decision Tree, kNN, and SVM. With this ensemble method the achieved accuracy, sensitivity, and specificity in prediction of the liver cancer are 88.4%, 88.4%, and 91.6%, respectively.”
- Formula for F1 score in equation (12) is incorrect. Please check.
Answer: The authors agree with the reviewer’s comment. The formula F1 was inspected and corrected.
- Future directions can be strengthened and limitations identified.
Answer: The future directions are strengthened and the limitations of this approach are identified in the last paragraph of the conclusion section. Citing from the last paragraph of the Conclusion section: “In future investigations, the original dataset will be enlarged if possible. The idea is to achieve an equal number of samples through all the dataset classes. The enlarged dataset could be a good starting point to develop a system of symbolic expressions for detecting Hepatitis C patients and the possible disease progression. Further investigation and tuning of GPSC hyperparameters are required to obtain very small symbolic expressions that have high classification accuracy. One of the limitations of this approach is the parsimony coefficient which has a large impact on the performance of GPSC. The influence of this hyperparameter will be reduced by ensuring that the dataset used is balanced and that correlation between variables is reasonably high.”
- Requires better proof-reading to correct language errors like “is two show,” “However The main …”, “poorly imbalanced dataset” etc.
Answer: The manuscript was proofread and the previously mentioned errors are corrected.
Reviewer 2 Report
In Andelic et al. authors make use of an evolution of genetic algorithm to detect hepatitis C patients and describe the disease progression. This work contributes with the calculation of simple mathematical equations that apparently are less computational expensive than current ML approaches. The study is well explained, structured and methods are sound regarding the questions to be answered. However, this reviewer has some concerns about the manuscript in its current state:
1. I am not convinced about the novelty or real contribution to domain of this approach. There are many ML interpreters that would do the same symbolic performance as the one presented in the manuscript (i.e., XGBoost + Deepliftshap, SHAP, etc.)
2. This reviewer is not become familiar with the underlying philosophy/style of the journal, but the article does not seem to be a regular scientific one, but a tutorial/exercise instead.
3. The work is based on the correlation assumption of variables, but authors after showing a table with negative results in linear correlation decided to use all the variables. What about Spearman's coefficient?
4. I do not understand lines from 330-333. Yet the number of PCs must be 12 not 10.
5. Please, when subpanels provided, use them as reference (i.e., Fig. 5, etc.)
6. Similar as 4. Authors decided to choose Bordeline SMOTE as oversampling method while other methods outperforms this one (Figs. 11/12).
7. The macro parameters in multiclass classification seems to be cherry-picked. Please more details are needed.
8. Authors also mention the average CPU execution time to discriminate amongst methods (Tables 5 and 6), but in all the cases that time is the same.
Author Response
The authors want to thank the reviewer for his time and effort to give constructive comments and suggestions which could greatly improve the manuscript's quality. The authors of this manuscript hope that the changes made in this manuscript will provide a suitable scientific contribution.
In Andelic et al. authors make use of an evolution of genetic algorithm to detect hepatitis C patients and describe the disease progression. This work contributes with the calculation of simple mathematical equations that apparently are less computational expensive than current ML approaches. The study is well explained, structured and methods are sound regarding the questions to be answered. However, this reviewer has some concerns about the manuscript in its current state:
- I am not convinced about the novelty or real contribution to domain of this approach. There are many ML interpreters that would do the same symbolic performance as the one presented in the manuscript (i.e., XGBoost + Deepliftshap, SHAP, etc.)
Answer: The authors agree that the given methods would probably achieve a similar performance as the one presented in the manuscript. However, this was not the idea of this manuscript. The idea was to see if using GPSC a symbolic expression can be obtained (equation) which could be used to detect HEPATITIS C patients from blood samples or to detect the progression of HEPATITIS C patients (Hepatitis C, Fibrosis, and Chirrosis). Those methods that were suggested by the reviewer would not produce the SYMBOLIC EXPRESSION (equation). After training these AI/ML methods the only thing that is obtained is trained AI/ML model.
To emphasize the idea once more the idea in this paper was to obtain symbolic expressions (mathematical equations) using GPSC algorithm which can be used to detect HEPATITIS C patients of HEPATITIS C disease progression. Again the idea was to obtain the mathematical equations !!!!!
Regarding the XGBoost algorithm, from our experience this algorithm produces excellent results on the training datasets i.e. the classification accuracy reaches 1.0. However, when new data is given to the trained algorithm the classification accuracy decreases. This types of algorithms generally produce higher standard deviation between train and test classification accuracy scores which could indicate potential overfitting. This standard deviation could be decreased by implementing 5-fold cross-validation but it still larger then the standard deviation generated with symbolic expression obtained with GPSC.
- This reviewer is not become familiar with the underlying philosophy/style of the journal, but the article does not seem to be a regular scientific one, but a tutorial/exercise instead.
Answer: The scientific contribution of this paper is to show how can highly imbalanced dataset be used in machine learning. Here the phrase “highly imbalance” indicates that for the fibrosis class there are only 12 samples in the original dataset. Besides balancing the novelty of this paper is to show how using GPSC the symbolic expression can be obtained which can be used for the detection of Hepatitis C patients or the progression of the disease using results obtained from blood samples.
To validate our approach the authors have implemented these equations on the original dataset and good classification accuracy was obtained.
The opinion of the authors is that this approach is novel in the field of Hepatitis C and the research presented in this paper can be of great importance to advance the detection of the disease using data obtained from blood samples.
- The work is based on the correlation assumption of variables, but authors after showing a table with negative results in linear correlation decided to use all the variables. What about Spearman's coefficient?
Answer: Regarding Pearson's correlation analysis any correlation (positive or negative) between input and output variables is good as long as it is in the range between -1.0 to -0.5 or 0.5 to 1.0 as it was described in the paragraph just before Pearson's correlation heatmap. If the correlation value is negative this means that if the value of the input variable increases the value of the output variable will decrease and vice versa. In the case of positive correlation if the value of the input variable increases the value of the output variable will also increase. The worst possible correlation is 0 which means that any change in the input variable value will not have any effect on the output variable value.
In Pearson's correlation heatmap, there are two input variables i.e. AST, BIL, and GGT that have the highest correlation values with the output variable (Category), and these values are 0.63, 0.55, and 0.42, respectively. Other input variables have low correlation values with the output variable in the range from -0.28 to 0.24. Two variables (sex, and ALP) have almost no correlation with the output variable (-0.04, 0.05).
Based on the shown Pearson's correlation heatmap authors concluded that generally correlation with the output (target) variable is bad and that the good approach would be to use all input variables in GPSC to generate symbolic expression. If some variables were omitted at this stage maybe it could influence GPSC performance and generate the symbolic expression with poor classification accuracy.
Regarding Spearman’s correlation heatmap the correlation values in this case slightly differ from the values of Pearson’s correlation heatmap.
- I do not understand lines from 330-333. Yet the number of PCs must be 12 not 10.
Answer: Since the other reviewer requested to use other non-linear dimensionality reduction techniques for better visualization of the data such as t-SNE or Kernel PCA these lines were omitted from the manuscript. In the revised version of the manuscript, the kernel PCA method was chosen to show the data graphically and with this method, the dimensionality reduction was performed from 12 to 2 variables.
- Please, when subpanels provided, use them as reference (i.e., Fig. 5, etc.)
Answer: In the revised version of the manuscript the Figure was renamed to Fig. and the Tables to Tab.
- Similar as 4. Authors decided to choose Bordeline SMOTE as oversampling method while other methods outperforms this one (Figs. 11/12).
Answer: The authors agree with the comment made by the second reviewer that it is odd that the authors have chosen symbolic expressions in binary and multiclass problems obtained on dataset balanced BorderlineSMOTE method. In both cases, the authors considered the classification accuracy and the length of obtained symbolic expressions. For example, if classification accuracy is higher than 0.9999 and the symbolic expression consists of 50000 elements it will not be considered the solution to the analyzed problem.
In the binary case, the length of the best symbolic expressions obtained on datasets balanced with Random oversampling, ADASYN, SMOTE, and BorderlineSMOTE are 91, 112, 423, and 60, respectively. The highest values of evaluation metrics were achieved in the case of SMOTE followed by ADASYN, BorderlineSMOTE, and Random Oversampling. The evaluation metric value in all cases for the binary problem is 0.985-0.999 the symbolic expression obtained in the case of the BorderlineSMOTE method was chosen for being the smallest symbolic expression that achieved high values of evaluation metrics when compared to the remaining three cases.
In the multiclass problem, the OneVsRestClassifier with GPSC as an estimator and the entire investigation was conducted using a random hyperparameter search method with 5-fold cross-validation. Since OneVsResClassifier was used and there are 4 classes then four symbolic expressions were obtained (each symbolic expression to detect each class). The lengths of symbolic expressions obtained in the case of Random Oversampling, ADASYN, SMOTE, and BorderlineSMOTE datasets are equal to 294/1961/998/2882; 696/90/675/216; 172/19/882/558; 417/148/471/118, respectively. In multiclass problems regarding the evaluation metric values sorted from highest classification scores to lowest are Random Oversampling, BorderlineSMOTE, ADASYN, and SMOTE. Symbolic expressions obtained on Random Oversampling are very large. For example, the symbolic expression used to detect cirrhosis patients has exactly 2882 elements. The evaluation metric values are slightly higher than those obtained in the case of the BorderlineSMOTE dataset. The symbolic expressions obtained in the case of ADASYN and SMOTE datasets were not considered due to low classification accuracy values. So, the authors have chosen the symbolic expressions obtained with the BorderlineSMOTE dataset since these symbolic expressions achieved really high classification accuracy with not-so-large symbolic expressions when compared to Random Oversampling case.
To conclude selection process of the best symbolic expressions in binary and multiclass problem were based not only on the evaluation metric values achieved but also on the size of the symbolic expression. In binary and multiclass problems the best symbolic expressions were obtained with a dataset balanced with the BorderlineSMOTE method.
- The macro parameters in multiclass classification seems to be cherry-picked. Please more details are needed.
Answer: The macro option of AUC, Precision, Recall, and F1-score were used since by definition the macro option calculates metrics for each class, and finds their unweighted mean. This does not take label imbalance into account. Since all dataset variations in a multiclass problem were balanced i.e. the number of samples in each class is equal this was the logical choice.
- Authors also mention the average CPU execution time to discriminate amongst methods (Tables 5 and 6), but in all the cases that time is the same.
Answer: Generally a major influence on execution time was the dataset size and the following GPSC hyperparameters: population size, number of generations, and parsimony coefficient. However, the dataset size is the same for all dataset variations so the dataset did not have any major influence on CPU execution time (average CPU execution time is the same in all cases when executing with the same GPSC hyperparameters). Each GPSC with a random hyperparameter search method and 5-fold cross-validation was executed until the termination criteria were satisfied. In some cases, the executions were faster and in some slower due to the bloat phenomenon (rapid growth of population members size without any significant decrease in fitness function value).
When average CPU time was calculated the average time of each fold in 5-fold cross-validation was measured as well as the average time of final evaluation.
As explained in the paragraph after Table 4. for the binary case is calculated as:
- average CPU time for training and validation on each split in 5-fold cross-validation is 20 [min]. Since there are 5-fold the total average CPU time was 100 [min] regardless of the dataset variation.
- average CPU time for final evaluation is an additional 20 [min].
To calculate the average CPU time the formula is:
20[min] * 5 + 20[min] = 100 + 20 [min] = 120 [min]
In a multiclass problem the procedure of calculating the total average CPU time is divided into the following steps:
- average time to perform one 5-fold cross-validation was 100 [min]
- since there are 4 classes using OneVsRest classifier the 5-fold cross-validation has to be executed 4 times (each time for a different class). This means there is a total of 20 GPSC executions (4 classes * 5-fold cross-validation on each class). Since the average CPU time for each split in 5-fold cross-validation is 20[min] this is equal to 400 [min].
20 [min] * 4 * 5 = 20 * 20 = 400 [min]
- The final evaluation is an additional 80 [min] of average CPU time since the OneVSRest classifier with GPSC estimator is trained on the train part of the dataset.
20[min] * 4 = 80 [min]
Total average CPU time = 20 * 4 * 5 + 20 * 4 = 480 [min] regardless of dataset variation used.
Round 2
Reviewer 1 Report
Review comments have been sufficiently addressed.
Reviewer 2 Report
Now, authors have clarified and improved the manuscript according to this reviewer's suggestions. I feel confortable with the idea of publishing the current form of the study in this journal!